# Dissociable encoding of evolving beliefs and momentary belief updates in distinct neural decision signals

Elisabet Parés-Pujolràs [1] ✉, Simon P. Kelly [1,3] & Peter R. Murphy [2,3]

Making accurate decisions in noisy environments requires integrating evidence over time. Studies of simple perceptual decisions in static environments have identified two human neurophysiological signals that evolve with similar integration dynamics, with one - the centroparietal positivity - appearing to compute the running integral and continuously feed it to the other - motor beta lateralisation. However, it remains unknown whether and how these signals serve more distinct functional roles in more complex scenarios. Here, we use a volatile expanded judgement task that dissociates raw sensory information, belief updates, and the evolving belief itself. We find that motor beta lateralisation traces the evolving belief across stimuli, while the centroparietal positivity locally encodes the belief updates associated with each individual stimulus. These results suggest a flexible computational hierarchy where context-dependent belief updates can be computed sample-by-sample at an intermediate processing level to modify downstream belief representations for protracted decisions about discrete stimuli.

Perceptual decisions are widely thought to involve a core process of evidence accumulation[1–3]. Extensive research across species has described multiple signals in distributed brain areas that are related to decision variables (DV), reflecting the evolving evidence tally during such accumulation (e.g., refs. 4–6 see ref. 7 for a review). This multiplicity of neural DV correlates has arisen through research predominantly using simple tasks that require the minimum computational ingredients of encoding and accumulating a single sensory evidence feature, usually in a single, contiguous evaluation of a stationary stimulus. Researchers are naturally turning to the question of what distinct functional roles these distributed signals might serve, and there is increasing recognition that answers may be found in employing tasks with more complex structure, context dependencies and/or integration strategies[8]. Here, we employ a volatile expanded judgement task in which individual sensory samples must be non-linearly transformed into appropriate belief updates, to investigate the potentially distinct computational roles of two human neurophysiological signatures of decision formation that normally exhibit similar dynamics during standard stationary-stimulus tasks: motor beta lateralisation (MBL) and the centroparietal positivity (CPP).

Beta-band activity (13–30 Hz) over the motor cortex is a well-known marker of motor preparation. Beta power gradually decreases during preparation of a movement of a contralateral limb[9,10], and typically reaches a stereotyped level just before action consistent with a threshold-crossing event[11,12]. In perceptual decision tasks, MBL emerges at a rate proportional to the strength of evidence, and when choices are freely reported with a manual response, peaks at the time of that response[11,13,14]. In delayed-response tasks, provided the stimulus-response mapping is known, MBL traces decision formation during evidence viewing and sustains even after commitment is reached, until the time of a subsequent response cue[15]. Further, in a decision-making context with volatility in the evidence source, MBL has been shown to approximate a DV computed through a non-linear accumulation process that is normative for this context[16].

The CPP, on the other hand, is a positive-going event-related potential (ERP) that evolves with the same evidence-dependent, build-to-peak accumulation dynamics as MBL during decision-making tasks

[1]School of Electrical and Electronic Engineering, University College Dublin, Dublin, Ireland. [2]Department of Psychology, Maynooth University, Co. Kildare, Ireland. [3]These authors jointly supervised this work: Simon P. Kelly, Peter R. Murphy. ✉e-mail: elisabet.pares-pujolras@ucd.ie

with continuous sensory evidence streams for any sensory modality[11], and slightly in advance of motor preparation[17]. Previous research has already established some notable properties in the CPP that are distinct from motor preparation signals: it traces decision formation even when no motor response is required[11], and in delayed-response tasks, peaks and falls at the time of choice commitment rather than sustaining until response, regardless of whether the response mapping is known during presentation of the sensory evidence[15]. Furthermore, where strategic adjustments and biases are enacted through changes in motor preparation starting levels, for example, related to prior knowledge of stimulus probability, response deadline, or evidence strength, the CPP undergoes a corresponding shift not in its starting level but rather in the amplitude it reaches at response[12,18].

The characteristics of these two signals are consistent with the view that the CPP reflects an accumulation process that is directly and continuously fed to the motor system, which executes no further computations on the sensory evidence but is subject to additional task-dependent biases and triggers a response once it reaches a stereotyped threshold level. However, since most previous studies have used continuous stimuli with stationary statistics and short integration times[11,12,15,18], it remains unclear whether and how the roles of the CPP and MBL might dissociate in more complex tasks that require additional computational steps and/or integration over longer time scales.

Here, we used a token-based sequential sampling task first developed in previous work[16] where the generative state of the task can change unpredictably during the formation of single decisions. This task has three key advantages for our purposes. First, the temporal separation between tokens (0.4 s stimulus onset asynchrony) allows us to precisely compute how the theoretical DV changes after each token, a process we refer to here as 'belief updating', and to directly map related quantities onto neural responses[16,19–21]. This contrasts with other frequently used decision-making paradigms (e.g., random dot kinematogram, RDK), where stimulus fluctuations are often too noisy and too fast to enable a precise link with computational variables and evoked potentials[22]. Secondly, our token task unfolds over longer timescales than traditional paradigms (i.e., trials lasting several seconds), allowing us to dissociate transient activities evoked by individual stimuli (i.e., responses to a particular sample) from sustained patterns of information encoding (i.e., an evolving DV that integrates over samples). Finally, the CPP has typically been studied in stationary (unchanging, non-volatile) environments where the optimal strategy is to weigh each piece of evidence equally and to integrate perfectly over time. In such scenarios, the information provided by each sample (objective evidence), and the change in the DV evoked by it (what we refer to here as effective evidence) are identical. In volatile environments, instead, these two quantities can be dissociated. According to the normative model for evidence accumulation on our task[23], for example, a sample providing strong evidence against an observer's current belief results in a bigger belief update (larger effective evidence) than a sample providing equally strong evidence for the observer's current belief[16]. Our task thus disentangles objective from effective evidence and allows us to test which of them may be captured by the CPP.

To preface our results, we find that centroparietal signals in this task context transiently encode the magnitude of belief updates evoked by each sample (effective evidence), but do not exhibit a longer-lasting, sustained component that tracks the DV over the course of the trial. By contrast, MBL reflects the evolving DV throughout the trial. We further show that single-trial fluctuations in both signals capture variability in behavioural choice over and above that accounted for by the sensory information on each trial and that variability in the CPP's encoding of belief updates are directly linked to residual fluctuations in MBL's encoding of a decision variable. Our study provides evidence for a key functional dissociation between these well-studied decision signals in the task setting investigated here,

and points toward a potentially flexible neural architecture for decision formation that may be reconfigured to meet the demands of different decision-making contexts.

## Results

### Task & behaviour

In our task ($N = 20$, Fig. 1A), participants monitored a set of checkerboard patches ('samples') appearing anywhere along a semicircular arc in the lower visual hemifield and centred on a central fixation point. Sample locations (specifically, polar angles relative to the fixation point) were drawn from one of two overlapping Gaussian distributions (generative states) with equal standard deviations, and means placed symmetrically to the left and right of the vertical meridian. Samples appearing close to the vertical meridian provided weak evidence for one or the other generative states, as they could be drawn from either distribution with similar probabilities. In contrast, samples appearing closer to the horizontal meridian provided strong evidence, since they were a lot more likely to be drawn from one of the two distributions than the other. Sample onset asynchrony was 0.4 s. Participants viewed a maximum of 10 samples per trial, and the distribution from which sample locations were drawn could switch during the trial with a fixed probability (hazard rate, H) of 0.1. Thus, some trials contained one, some more than one, and the rest contained no changes in the generative distribution (Fig. 1A, right). The participants' task was to report what they believed to be the generative state (i.e., the 'active' distribution) at the end of each trial. They completed 18 blocks of 80 trials across 2 sessions.

The normative model for this task prescribes that participants transform their posterior beliefs after accumulating each sample to form the prior for the next sample, according to a non-linear function that depends on their estimate of the environmental volatility (Fig. 1B; see Eqs. 1, 2 in Methods). In effect, this non-linearity limits the degree of belief that can be attained for either alternative and facilitates fast revision of that belief following state changes (Fig. 1C, upper panel), particularly when the volatility is estimated to be high. Our participants performed the task better ($83.8\% \pm 0.66\%$; mean ± s.e.m.) than expected if they followed simple strategies such as of choosing based on the last sample only ($79.7\% \pm 0.14\%$; $p < 0.001$) or the sum of all available samples (i.e., perfect accumulation, $75.5\% \pm 0.22\%$; $p < 0.001$), but worse than the ideal observer employing the normative updating rule with perfect knowledge of the generative statistics and no noise or bias ($87.5\% \pm 0.20\%$; $p < 0.001$; Fig. 1D).

The non-linear transformation of the evolving belief, that is the key feature of the normative approach to solving this task, has a number of behavioural consequences. First, its slope is always <1, which produces leaky accumulation and a recency effect in evidence weighting. Further, it produces choice behaviour that is particularly sensitive to samples indicating that a potential change in the generative state has occurred (i.e., samples that clearly conflict with a strongly held current belief), as well as to samples encountered when the belief is generally weak (i.e., when uncertainty about the current generative state is high[16]; Fig. 1C). To test whether participants exhibited these behavioural hallmarks of normative belief updating, we fit logistic regression models estimating the average weight on choice of each sample in the trial sequence (Eq. 7 in Methods). The evidence provided by each sample was quantified in terms of its log-likelihood ratio (LLR). Further, we computed the surprise associated with a sample (defined here as the analytically computed change-point probability, which quantifies the likelihood, given the existing belief, that a sample directly followed a change in generative state; see Eqs. 4–6), and the uncertainty in the belief at the time of encountering that sample (see Methods). As predicted by the normative model, we found that participants weighted evidence occurring later in the trial more heavily ($p < 0.001$, two-tailed permutation test on weights for the first 5 versus the last 5 samples), and also upweighted samples associated

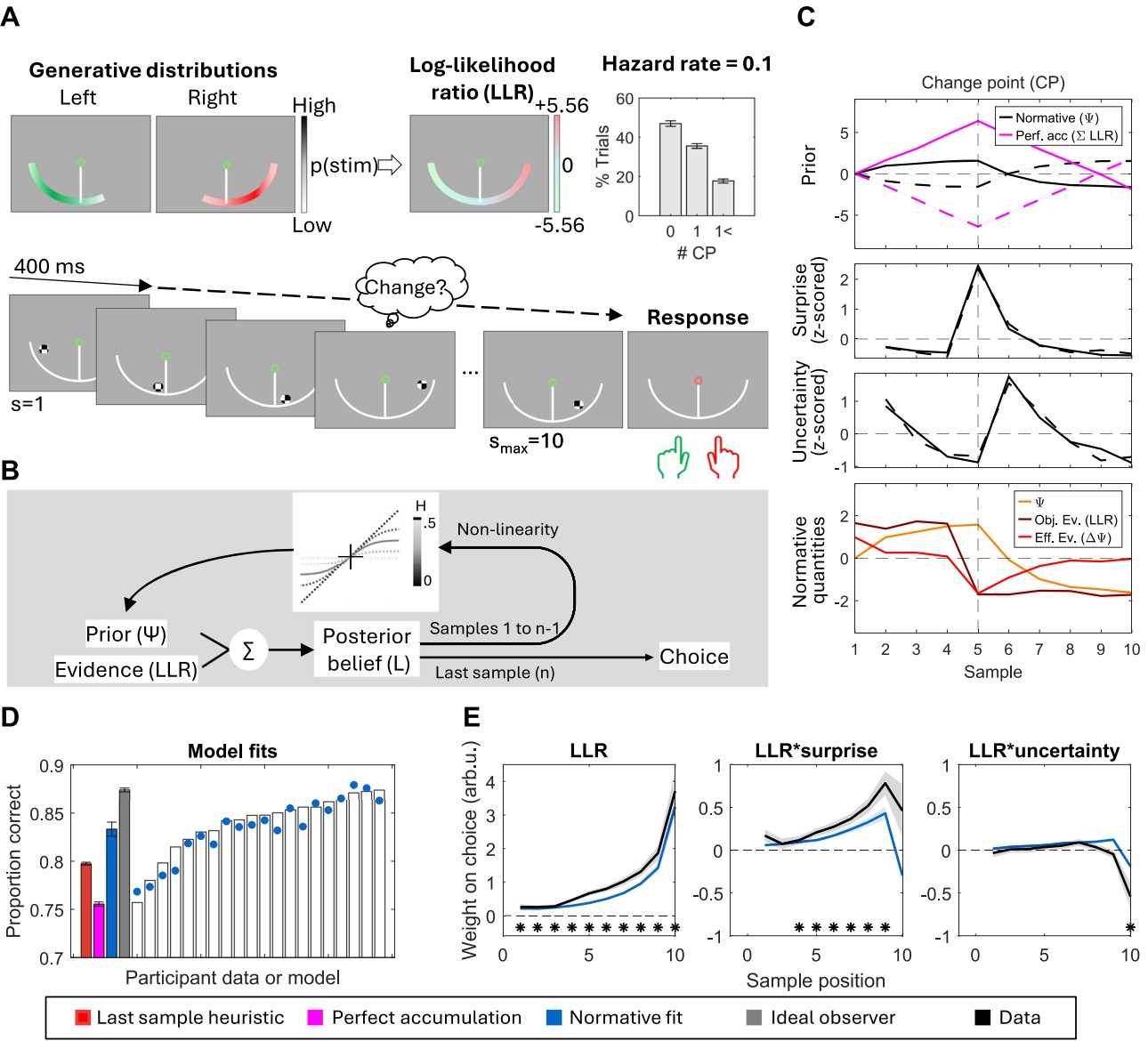

**Fig. 1 | Task and behavioural results. A** Participants monitored a sequence of samples of fixed duration (max. 10 samples). Upon presentation of a response cue, they reported which generative distribution (left/right) was active at the end of the trial by pressing a key. There was a 0.1 probability that the generative distribution might change after any given sample (Hazard rate, $H = 0.1$), so in most trials, there were either no change points (CP) or only 1 (see inset). **B** Normative belief updates schematic, including the non-linearity across various levels of volatility (H). The solid grey line illustrates the non-linearity at $H = 0.1$. **C** Time course of prior (top), surprise (middle) and uncertainty (bottom) over samples for an illustrative example of simulated trials ($N = 100$) with a single CP at sample 5. Generative sources for samples in these trials switched from left to right (dashed line) or right to left (solid line). An ideal observer's normative decision variable, tracked by the normative prior ($\Psi$), plateaus shortly after the trial starts, and rapidly changes signs after the change in the generative distribution. In contrast, a perfect accumulator grows linearly over time and requires more evidence to flip sign after a CP (magenta). Surprise values (i.e., change point probabilities) peak at the time of a CP, and uncertainty values (-|$\Psi$|) decrease as priors grow. The bottom panel illustrates how normative beliefs ($\Psi$) and step-wise belief updates ($\Delta\Psi$) evolve in this subset of trials. While the mean LLR values (objective evidence) remain stable throughout the duration of any given state (i.e., before and after sample 5 in this example), belief updates become smaller (effective evidence, $\Delta\Psi$) as beliefs grow stronger. **D** Proportion of observed and model-predicted correct responses. White bars and overlaid coloured points represent individual data and fits, respectively. Filled bars display grand-averaged data ± s.e.m. **E** Observed and model-predicted (mean ± s.e.m.) weight on the choice of each sample's LLR, and its interaction with change point probability (surprise) and uncertainty. Asterisks along the bottom indicate samples where weighting significantly differed from zero ($p < 0.05$, two-tailed cluster permutation test).

with high change-point probability ($p < 0.001$, two-tailed cluster permutation test against zero; Fig. 1E). By contrast, we did not find a reliable positive modulation by uncertainty ($p = 0.286$ positive effect at sample 7). This null effect is in contrast to previous data from similar versions of our task[16], though a weaker effect in the present case is consistent with normative model predictions due to differences in task generative statistics (here, a higher H and higher signal-to-noise in generative distributions; see Methods). Note also that "surprise" and

"uncertainty" are not quantities directly used in the normative computations, but rather sensitivities to these variables are diagnostic of the normative belief updating strategy that we can test for in our data. The normative model was able to account for these behavioural effects (Fig. 1E) with estimated participant-specific subjective hazard rates (H) that did not differ from the true generative $H = 0.1$ at the group level ($M = 0.092$, s.e.m. = 0.015; one sample $t$ test, $t_{(19)} = -0.54$, $p = 0.59$, d = 0.12, 95% CI = [0.05, 0.12]). It also fits the data better than several

alternative models (see Supplementary Fig. S1 for full parameter estimates and model comparison details).

Thus, consistent with previous work using the same task[16], our results show that participants' behaviour approximated normative belief updating. This, in turn, allowed us to use the fits of the normative model to derive decision-relevant quantities with which to investigate our neural signals of interest: MBL and the CPP.

## Motor preparation signals track the ongoing state of a decision variable

We first investigated motor preparation signals. Beta power over the motor cortex progressively lateralized during stimulus viewing, indicating increasing motor preparation and commitment to a specific action plan as the trial unfolded (Fig. 2A). In trials where changes in the generative distribution occurred, the time at which the response-specific lateralisation became apparent varied with the time of the last change point (Fig. 2B).

To test the sensitivity of MBL signals to evidence- and decision-related quantities from the normative model fits, we segmented the motor beta lateralisation traces [Left-Right hemispheres] around each evidence sample s (see Methods) and conducted time-resolved regressions against a set of model-derived variables. In our first analysis (Eq. 9), we included the prior belief ($\Psi_s$), the evidence provided by the new sample (LLR), and its modulation by surprise (LLR*surprise) - terms that linearly sum to approximate the updated belief ($\Psi_{s+1}$) subject to the non-linear normative transformation as observed in our data[16]. Uncertainty modulations were not included in EEG analysis due to their lack of a reliable behavioural effect in our dataset. We constructed the regression models such that positive coefficients would indicate lateralisation for the response favoured by the normative model given the evidence. Based on previous work (with MEG rather than EEG) suggesting that MBL encodes the evolving belief during trials of our task ([16], we expected a specific pattern of effects across the three regressors: sustained prior ($\Psi_s$) coefficients across the entire sample epoch, indicative of stable encoding of the belief before presentation of the new sample; and a sample-evoked increase in the coefficients for each of the individual sample-related terms (LLR and its modulation by surprise) which, as described above, would linearly sum with prior encoding to reflect the updated, normatively transformed belief after accumulating the new sample[16]. By contrast, a signal encoding the objective sensory evidence carried by each sample would display a reliable effect only for the LLR term in this regression model; whereas a signal encoding only the effective evidence for the current sample (that is, the magnitude of the belief update but not the overall belief) would show reliable effects for LLR and its modulations, but not the prior term.

Our analysis showed that MBL followed the pattern expected of a signal encoding the evolving belief. It showed reliable encoding of the model-estimated prior belief ($\Psi$) that was sustained across the entirety of the analysed sample epoch (Fig. 2C, D; $p < 0.001$). Further, MBL showed an evoked lateralisation following each sample, the magnitude of which scaled with each sample's LLR ($p < 0.001$) and was further modulated by how surprising it was ($p = 0.002$).

To maintain consistency with our analyses of the CPP signal below (see Methods), we also ran a simplified regression including only prior belief ($\Psi_s$) and effective evidence ($|\Delta\Psi| = \Psi_{s+1} - \Psi_s$) as regressors (Eq. 10) – with the latter term capturing the full extent of sample-by-sample normative belief update as compared to its approximation by LLR with surprise modulation described above. This model similarly yielded both the sustained prior ($\Psi_s$) and post-sample belief update ($\Delta\Psi$) encoding of an approximately normative DV (all $p < 0.001$; Fig. 2E), and provided a comparable fit to the data (paired $t$ test of adjusted R squared values, $t_{(19)} = 0.74$; $p = 0.467$; $d = 0.165$; 95% CI = $[-1.04*10^{-4}, 2.18*10^{-4}]$). To confirm that MBL tracked the normative DV across samples better than it tracked momentary stimulus

information, we compared the adjusted $R^2$ values of regressions that each included a single regressor: a pure decision variable (DV, $\Psi_{s+1}$), the effective evidence (EE, $\Delta\Psi$) or objective evidence (OE, LLR). MBL was indeed best described by a regression encoding the state of an evolving belief (Fig. 2F).

In sum, our results showed that motor beta lateralisation approximated a normative DV, with a sustained pattern of activity reflecting integrated information across samples as well as sample-evoked responses that are indicative of a normative belief updating process. These evoked lateralisation activities resulted from a decrease in power over the hemisphere contralateral to the response supported by the evidence, and an increase in power over the ipsilateral one (Supplementary Fig. S2).

## Centroparietal potentials reflect normative belief updates

We next similarly tested for distinct slow-building (across samples) and transient (sample-evoked) components of evidence processing present in the CPP. We observed that the centroparietal ERP was predominantly characterised by transient positive-going responses to each individual evidence sample rather than protracted build-up over the duration of the trial (Fig. 3A). Although some channels did show a positive drift over the whole trial, these positive deflections were not consistent across neighbouring electrodes in comparison with the transient sample-evoked responses. In order to examine transient and slow-building signal components systematically, we conducted time-resolved regressions of ERP amplitude against relevant decision-related quantities. Following the same logic as for motor beta lateralisation, in our first analysis (Eq. 11), we included the model-estimated prior belief ($|\Psi|$), the evidence provided by each sample ($|LLR|$), and its modulation by surprise ($|LLR|*$surprise). In this case, we took the absolute (i.e., unsigned) values of the prior and LLR to account for the fact that the CPP is positive-going in response to sensory evidence regardless of the decision alternative that it favours. To allow for slow-building effects over trial time, we baseline-corrected the data for this analysis prior to trial onset and employed a high-pass filter with a low cutoff frequency (0.1 Hz). If a slow-building positivity encodes the overall DV across samples, we would expect to find a sustained, positively weighted sensitivity to the absolute prior belief $|\Psi|$, similar to the encoding of the signed prior observed in motor beta lateralisation (see Fig. 2).

We found that centroparietal responses to each sample scaled with its $|LLR|$ ($p = 0.007$, two-tailed cluster permutation), and were further modulated by how surprising the sample was ($p = 0.017$, two-tailed cluster permutation, Fig. 3B; Eq. 11). Conversely, we found no evidence for encoding of the unsigned prior (no positive clusters identified). Given that some channels showed a positive drift over trial time, we repeated the analysis on all 128 scalp electrodes without averaging over channels, but no significant clusters encoding the unsigned prior were found anywhere on the scalp ($p > 0.05$; two-tailed cluster permutation test; Fig. 3B). We further repeated the analysis in various subsets of channels, including those that exhibited the strongest positive ramping activity over trial time, and no |prior| effect emerged either (see Supplementary Fig. S4).

The modulation by surprise indicates that for a given level of $|LLR|$, the sample-evoked CPP is enhanced for more surprising samples and relatively reduced for unsurprising ones. This suggests encoding not of the objective evidence carried by each sample per se (i.e., its $|LLR|$), but rather the effective evidence that determines the belief update evoked by each sample after, in the normative model, it is accumulated and the belief passed through the normative non-linearity (see Eq. 12: $|\Delta\Psi| = |\Psi_{s+1} - \Psi_s|$). Supporting this interpretation, a simpler regression including only the magnitude of the belief update $|\Delta\Psi|$ revealed significant encoding of this variable over the same group of centroparietal electrodes ($p = 0.008$; Fig. 3C).

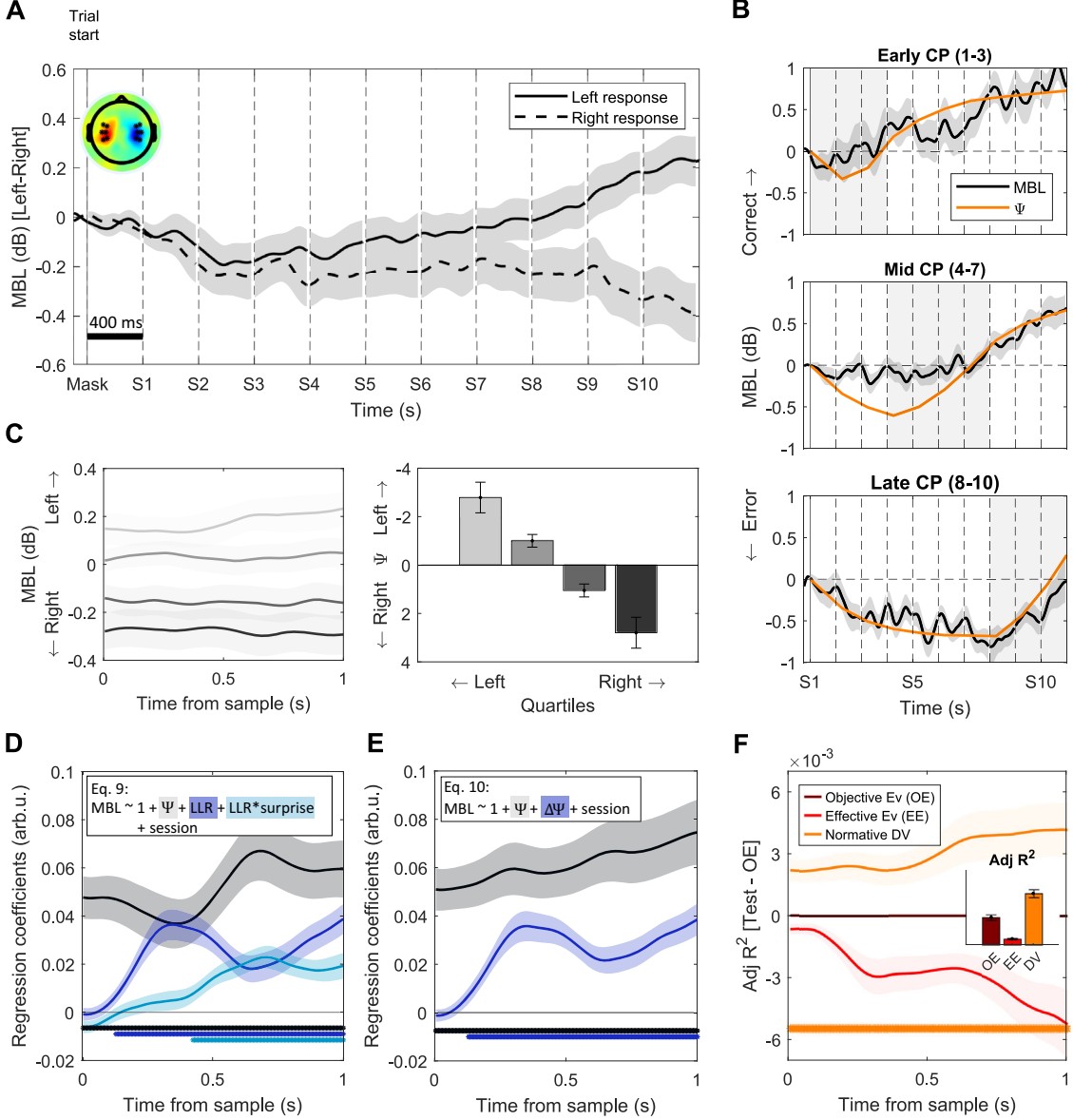

**Fig. 2 | Motor beta power lateralisation encodes the evolving decision variable.**
**A** Grand-averaged (± s.e.m.) beta lateralisation over the motor cortex [left-right hemispheres] aligned to trial start, sorted by response. Topography illustrates [Left-Right] hand responses at the end of the trial [4.2–4.4 s]. **B** Grand-averaged (± s.e.m.) relative motor preparatory activity measured as the difference in MBL [Left-Right hemispheres] between left- and right-hand responses for trials containing a single change-point (CP) at different latencies. Positive values indicate increasing motor preparation towards the final response. Overlaid orange traces indicate the estimated normative prior (Ψ, scaled for visualisation) for the same set of trials, and illustrate that MBL closely mirrored the evolution of a normative DV over trial time. For illustration purposes, only correct trials are included. **C** Grand-averaged (± s.e.m.) motor beta lateralisation aligned to sample onset and averaged over samples 2–10 (left) and split in quartiles according to the belief (Ψ) at sample onset (right). **D** Standardised regression coefficients (mean ± s.e.m.) of the motor lateralisation [Left - Right] activity resulting from the regression, including prior

($p < 0.001$), LLR ($p < 0.001$), and the modulation of LLR encoding by surprise ($p = 0.002$) (Eq. 9), averaged over samples 2-10 and aligned to sample onset.
**E** Standardised regression coefficients (mean ±s.e.m.) of the motor lateralisation resulting from the regression with only prior and effective evidence (all $p < 0.001$) factors (Eq. 10), averaged over samples 2−10 and aligned to sample onset. **F** Single-regressor model comparison. Traces indicate the difference in adjusted $R^2$ values for regressions including only the updated normative decision variable (DV, $Ψ_{s+1}$) or only effective evidence (EE, $ΔΨ$) as regressors, minus a regression including only absolute objective evidence (OE, LLR). The adjusted $R^2$ of the OE regression is used as a reference and appears as a flat line with value 0 in the plot. The normative DV regression explained significantly more MBL variance than the objective evidence one ($p = 0.002$). Bar graphs indicate Adj. $R^2$ values for each regression (mean ± s.e.m.), across the whole sample epoch. In all panels, asterisks indicate significant cluster periods (two-tailed cluster-based permutation test, $p < 0.05$).

Finally, to confirm that a model where centroparietal responses encode effective evidence provides a better fit relative to other candidates, we compared the adjusted R-squared values of this last regression (effective evidence, $|ΔΨ|$) to those of a regression with $|LLR|$ as the only regressor (objective evidence) and those of a regression with the updated decision variable ($|ΔΨ|_{s+1}$). The effective evidence regression yielded significantly better fits for a

centroparietal cluster of electrodes consistent with the topographic peak of the CPP ($p = 0.016$; Fig. 3D). All these results remained significant in a control analysis where we excluded all samples where saccades or microsaccades were detected (Supplementary Fig. S5). Comparing subsets of trials with equally high values of $|LLR|$ but that differed in effective evidence, illustrates that the response to high $|ΔΨ|$ was greater than to low $|ΔΨ|$ (Fig, 3E), showing how the CPP's

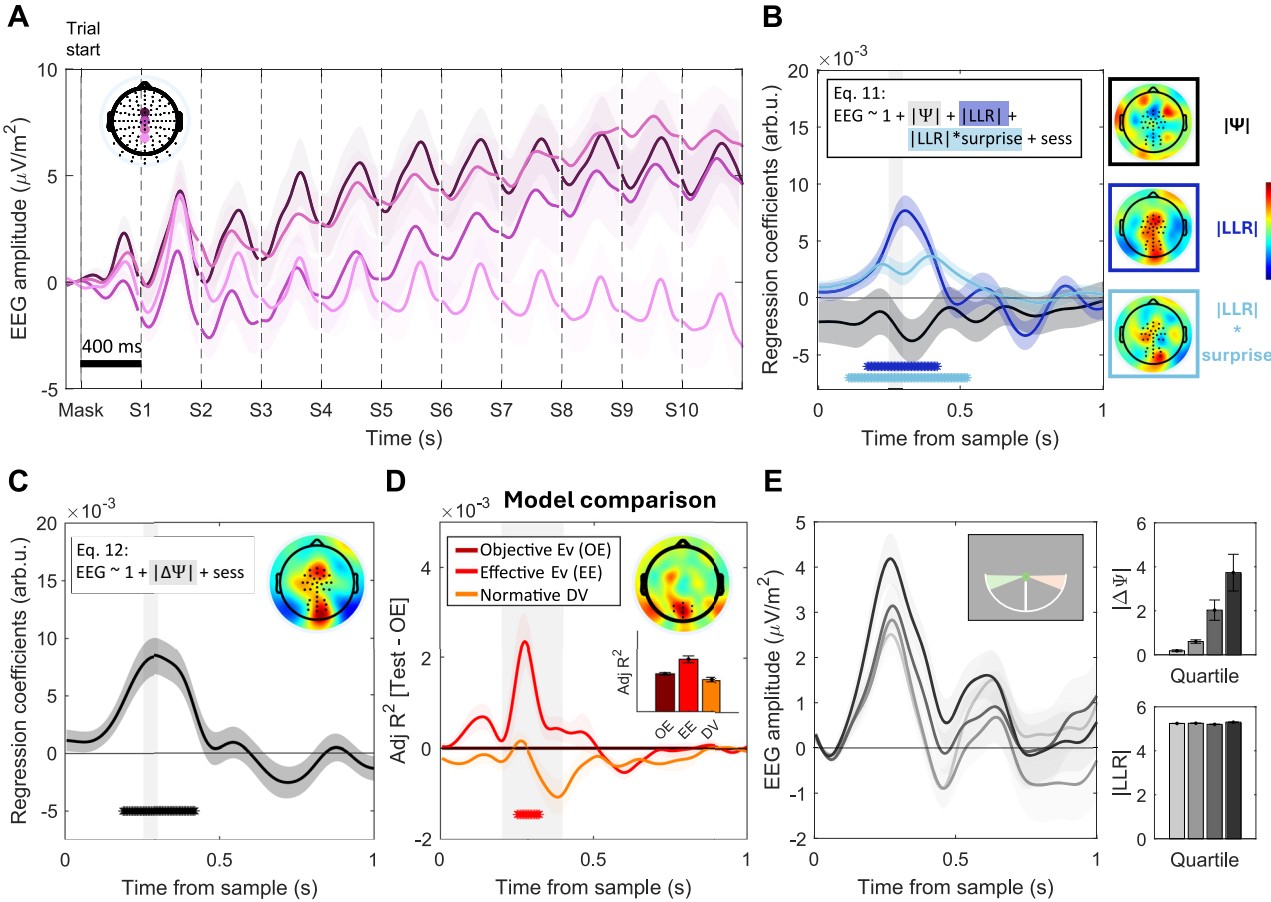

**Fig. 3 | Centroparietal signals reflect normative belief updates. A** Grand-averaged ( ± s.e.m.) ERP traces aligned to trial onset, with baseline correction by the period preceding the trial onset, for the highlighted electrodes. **B**, **C** Standardised regression coefficients (mean ±s.e.m.) for a regression including $|\Psi|$, $|LLR|$ ($p = 0.007$) and the $|LLR|$ modulation by surprise ($p = 0.017$). (Eq. 11, **B**) or a simplified version including only effective evidence ($|\Delta\Psi|$) as a regressor ($p = 0.008$). (Eq. 12, **C**), averaged over samples 2–10 and aligned to sample onset. Topographies show whole-scalp regression coefficients [0.25-0.3 s] post-sample, averaged over samples 2-10. **D** Centroparietal activity is best explained by an effective evidence model. Traces indicate the difference in adjusted $R^2$ values (mean ± s.e.m.) for regressions including only the updated normative decision variable (DV, $|\Psi_{s+1}|$) or only effective evidence (EE, $|\Delta\Psi|$) as regressors, minus a regression including only absolute objective evidence (OE, $|LLR|$ )(see Methods). As in Fig. 2F, the adjusted $R^2$ of the OE regression is used as a reference and appears as a flat line with a value 0 in the plot. The effective evidence (EE) model was significantly better than the objective evidence (OE) one ($p = 0.016$). Topographies show the average difference in the regressions' adjusted $R^2$ [Adj. $R^2(|\Delta\Psi|)$ - Adj. $R^2(|LLR|)$] within the time window indicated by the shaded area, averaged over samples 2-10. Bar graphs indicate Adj. $R^2$ values for each regression (mean ± s.e.m.) within the time window indicated by the shaded area. **E** Standard (not regression-derived) grand-averaged ( ± s.e.m.) ERP traces aligned to trial onset restricted to samples with $|LLR| > 4.75$ (coloured areas in the top right inset) and split into quartiles according to the magnitude of the belief update (effective evidence) they evoked. Data were baselined at sample onset for illustration purposes. Samples appearing in the same absolute angular location (i.e., same objective evidence or $|LLR|$ ) evoked higher potentials when they resulted in a bigger normative belief update. Bar graphs (right) show the mean ( ± s.e.m.) $|\Delta\Psi|$ (top) and $|LLR|$ (bottom) of each quartile. In all panels, asterisks indicate significant cluster periods (two-tailed permutation cluster test, $p < 0.05$).

encoding of effective evidence is truly distinct from objective evidence ($|LLR|$ ). Note that even when the required belief update is close to 0 (lowest $|\Delta\Psi|$ quartile), it is expected that the CPP still shows an appreciable positive deflection: whereas noise in the underlying belief update computation may take either a positive or negative sign and would thus average out to 0 across trials, the CPP reflects the absolute value of this belief update quantity, and so the average still results in a positive-going potential. Supplementary analyses showed that the sensitivity of the CPP to effective evidence observed here cannot be accounted for by collinearity between this quantity and various forms of surprise (Supplementary Fig. S7). Alternative trial sorting heuristics that correlate with different average |effective evidence| values, such as grouping samples based on whether or not they occurred at a change point or as a function of their consistency with prior beliefs, showed similar effects (Supplementary Fig. S6).

## Linking centroparietal potentials, motor beta lateralisation and behaviour

We next sought to directly link fluctuations in our neural signals of interest to one another and to variability in choice behaviour. To do so, we exploited the trial-by-trial variability in the neural signals. First, we extracted the residual fluctuations in EEG activity following a given sample that could not be explained by computational variables derived from the normative model and investigated how these related to choice (schematic in Fig. 4A, C). In a separate analysis, we aimed to link the two neural signals at a single-trial level by testing whether trials where, for example, a particularly high centroparietal response was observed for a given combination of model variables, were accompanied by a particularly large belief update at the motor preparation level (Fig. 4E). In order to increase our signal to noise ratio for these single-trial analyses, we applied an additional high-pass filter (>1 Hz) to the EEG data prior to re-running the regressions in Eq. 12 (following

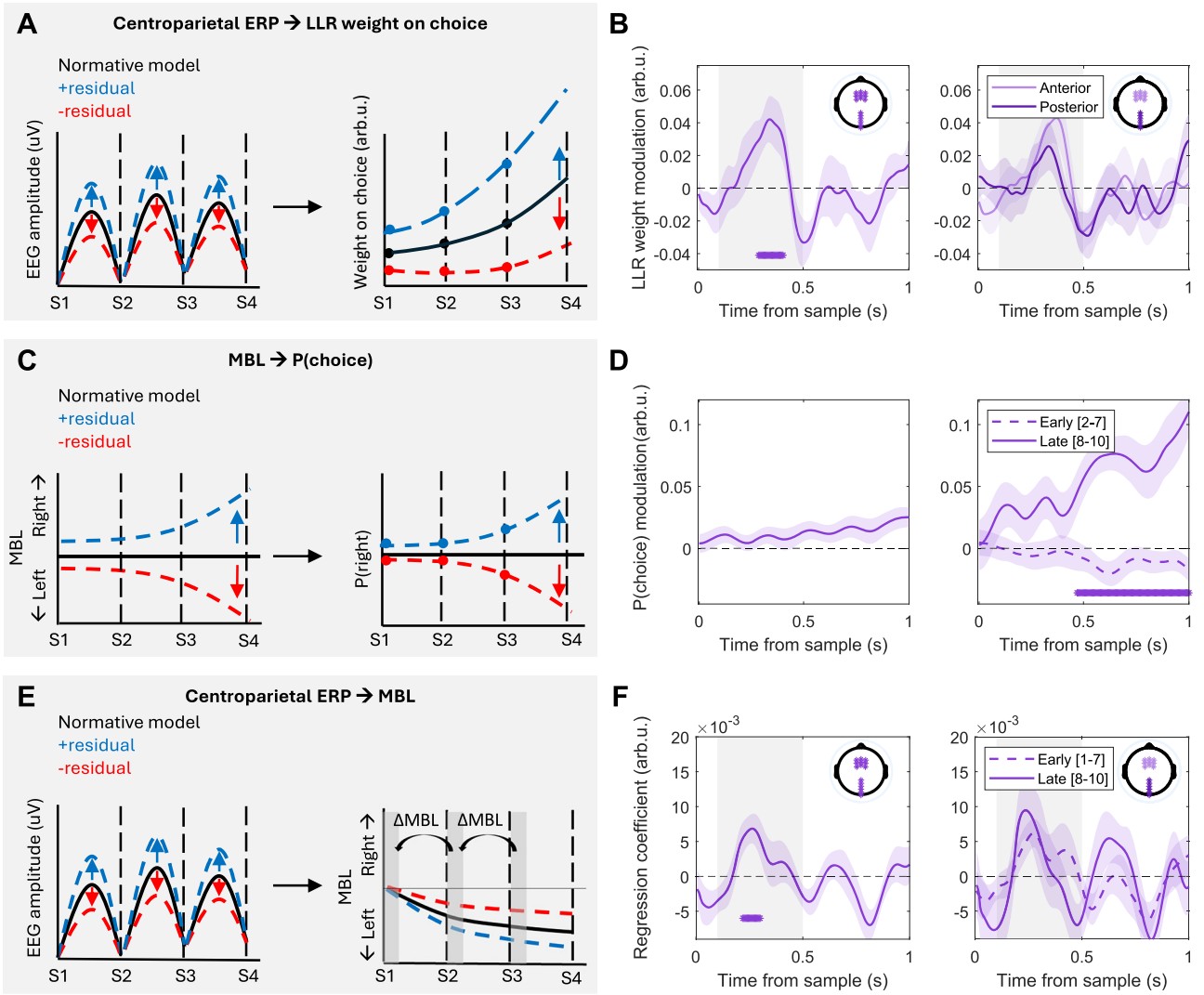

**Fig. 4 | Single-trial fluctuations link centroparietal signals, motor lateralisation and choice. A, B** Single trial fluctuations in centroparietal evoked responses modulate sample weight on choice. **A** Black lines represent hypothetical EEG traces & evidence weights on choice predicted by the normative model, and coloured lines represent positive (blue) or negative (red) residuals in Eq. 12 (left) and their predicted effect on evidence weight modulation (right). **B** Regression coefficients (mean ± s.e.m.) of Eq. 13, showing that trials where CPP potentials (in highlighted electrodes) were higher than predicted by the normative model (i.e., positive residuals from Eq. 12) were related to increased evidence weights ($p = 0.008$). **C, D** Single-trial fluctuations in motor lateralisation influence choice probabilities at the end of the trial. **C** Black lines represent the beta lateralisation & choice probabilities predicted by the normative model at any point in time (here taken as a baseline), and coloured lines represent positive (blue) or negative (red) residuals in Eq. 9 (left). These indicate lateralisation favouring right or left responses over and above any lateralisation predicted by the normative model, and their predicted

effect on choice probabilities (right). **D** Normalised regression coefficients (mean ± s.e.m.) of Eq. 14, showing that motor lateralisation residuals at the end of the trial influence choice probabilities ($p = 0.001$). **E, F** Centroparietal signals influence updates to the decision variable encoded in motor beta lateralisation. **E** Black lines represent hypothetical EEG traces/motor beta lateralisation, and coloured lines indicate positive (blue) or negative (red) residuals in Eq. 12/9, left/right respectively. **F** Standardised regression coefficients (mean ± s.e.m.) of Eq. 17, showing that variability in centroparietal responses influences motor beta lateralisation updates following each sample across all samples (left, $p = 0.025$), and for early and late sets of samples (right). Motor beta lateralisation updates (ΔMBL) were computed as the difference in beta lateralisation during the initial 100 ms in consecutive samples. In all panels, asterisks indicate significant cluster periods (two-tailed cluster-based permutation tests, $p < 0.05$). Where present, shaded areas indicate cluster test period; in all other panels, the full 1 s epoch was included.

previous work, see e.g., ref. 20), and extracted the residuals from a subset of anterior & posterior electrodes where effects were found to be maximal (see Methods & Supplementary Fig. S3).

Trial-by-trial variations in centroparietal responses modulated the weight that each sample had on choice ($p = 0.008$; two-tailed cluster-based permutation test; Fig. 4B, left). That is, samples that evoked larger-than-expected centroparietal responses for a given level of effective evidence (i.e., trials with positive residuals from Eq. 12) also had stronger weight on choices. No differences were observed between results in anterior compared to posterior clusters ($p > 0.05$; two-tailed cluster-based permutation test; Fig. 4B, right), respectively,

nor between early and late samples in the trial ($p > 0.05$; two-tailed cluster permutation early vs. late). In turn, residual fluctuations in motor preparatory activity occurring late in the trial influenced the overall choice probabilities (Fig. 4D, right; two-tailed cluster-based permutation test for early [2–7] vs. late samples [8–10], $p = 0.001$; the effect was not significant ($p = 0.08$) when averaging across all samples; Fig. 4D, left).

Finally, we aimed to directly link variability in the centroparietal encoding of belief updates to the evolving representation of the belief in motor beta lateralisation (Eq. 17). We hypothesised that, in the context of our task, the effect of single-trial CPP fluctuations on

evidence weighting reported above would be mediated by the motor signal that ultimately drives behaviour. Accordingly, we found that residual EEG fluctuations in centroparietal channels significantly modulated the sample-evoked change in the motor preparation signals. Namely, trials where the evoked centroparietal responses were larger than expected given the computational variables from the normative model fits (i.e., trials with positive residuals from Eq. 12) were also accompanied by greater-than-expected changes in lateralisation at the motor preparation level ($p = 0.025$; Fig. 4F, left). This effect was not significantly different between early and late samples (Fig. 4F, right; two-tailed cluster-based permutation test for early [1–7] vs. late samples [8–10]), nor did it differ between the two groups of electrodes (Supplementary Fig. S8).

Note that the time course of the modulatory effect of centroparietal residuals on evidence weighting (Fig. 4B), as well as its effect on motor lateralisation updates (Fig. 4F) matched that of the neural encoding of belief updates over centroparietal channels (see Fig. 3). The temporal specificity of these effects indicates that only residual fluctuations specifically in the encoding of belief updates in these signals have an impact on choice and motor preparation.

## Discussion

Making perceptual decisions about external stimuli often requires the integration of multiple pieces of information. While several brain signals have been linked to decision variables (DV) exhibiting integration dynamics during a contiguous evaluation of a stationary sensory feature, it is important to know whether and how functional roles dissociate in more complex tasks. In this study, our goal was to directly test which specific aspects of a belief updating process are encoded by two prominent human EEG signatures of evidence accumulation - the CPP and MBL - in a volatile expanded judgement task. Specifically, we investigated whether these signals tracked an evolving belief in a sustained manner across successive discrete evidence samples, or whether they reflected momentary sample information or the belief updates derived therefrom. We used a task that required participants to integrate evidence in a non-linear manner, divorcing the objective evidence carried by each sample from its effective influence on the DV. Further, the discrete, temporally separate evidence samples we used allowed us to directly map decisional quantities to evoked neural responses. We applied a recent formulation of a normative model that can account for how agents flexibly tune their accumulation strategies in environments with various degrees of volatility[16,23], and which allowed us to precisely quantify how a DV evolved over trial time, including the precise belief updates (i.e., changes in the DV) required after each sample in our task. This normative model provided a close fit to participants' behaviour, as expected from previous work[16], and thus we could use model-derived quantities to test which aspects of a decision might be encoded by the CPP and MBL. We found that (1) MBL consistently reflects the state of an evolving normative DV, (2) the CPP encodes normative belief updates but is not significantly associated with the overall state of the DV, and finally, (3) single-trial fluctuations in the CPP in this context can account for otherwise unexplained deviations from the normative DV in MBL signals, providing a direct link between the processes indexed by these two signals.

Our first finding replicates previous MEG finding[16] that MBL approximates the normative DV (i.e., the strength of beliefs) on our volatile decision-making task, with a sustained pattern of activity that reflects across-sample integration as well as discrete sample-evoked responses reflecting a normative belief update. The magnitude and sign of MBL at any given time during the trial reflected the DV, and updates were implemented by modulating power in both hemispheres, according to the sign of the belief update (i.e., simultaneous ipsilateral increase and contralateral decrease of power, as a function of the sign of the DV; see Supplementary Fig. S2). Next, we addressed the main question of how the CPP participates in this task, specifically,

whether slow, sustained components of the CPP encode a DV, and how fast centroparietal responses evoked by each sample relate to the information provided by that sample. In contrast to MBL, we found no significant sustained encoding of the DV across samples, but rather a transient encoding of an intermediary quantity, the normative belief update derived from each sample.

Previous work comparing the dynamics and functional properties of the CPP with motor preparation activity had employed simple tasks requiring a rapid, contiguous evaluation of a stationary, elemental stimulus feature (e.g., motion direction or stimulus contrast, without volatility in generative task state), where momentary evidence that is readily encoded in sensory cortex directly translates into appropriate increments in an accumulating DV. In those contexts, it was shown that the CPP exhibits many key characteristics of the accumulation process tracing the overall DV: it builds positively from a stimulus-aligned onset, at a rate proportional to evidence strength, up to a peak at the time of decision commitment, and its amplitude at that time varies with choice accuracy[2,11,12]. Apart from additional task-dependent biases and urgency components found to be added at the motor level in some contexts, the evidence-dependent dynamics exhibited in the CPP were mirrored in downstream motor preparation, at first blush suggesting a degree of computational redundancy. Our current study demonstrates that the signals' computational roles can diversify in tasks with discontinuous evidence samples requiring more computational steps to transform stimulus information into decision-relevant quantities. This gives new insight into the flexible cognitive capabilities afforded by the two-level architecture for decision formation reflected in the CPP and motor preparation signatures. Specifically, our results suggest that in task contexts where evidence comes in discrete samples, an intermediate process computes appropriate belief updates, which are then accumulated downstream at the motor level. In the advancing area of decision research in the past 3 decades, a commonly noted hallmark of the neural decision variable correlates that has distinguished them from the sensory encoding of the evidence itself is their "freedom from immediacy"[3], whereby they keep a running total of cumulative evidence for as long as it takes to arrive at an ultimate commitment. In this sense, the CPP in the context investigated here is revealed to occupy an interesting middle ground, not concerned with basic sensory processing but handling intermediate cognitive computations applied to sensory information in order to feed a more slowly evolving tally downstream. While previous MEG work[16] was able to trace the encoding of objective sensory evidence through visual cortical areas and, at the far end of the pathway, a motor plan reflecting the evolution of the normative decision variable across tokens, no correlates of the intermediate neural processing steps required to transform the former to the latter had been identified. Our finding that the CPP uniquely encodes this critical normatively-scaled quantity, therefore, adds a missing link to the chain of events leading from objective stimulus information to a normative decision variable.

Previous work had suggested that sample-evoked centroparietal signals can be better explained by momentary rather than cumulative evidence in expanded judgement tasks[20,21]. However, the possibility that trial-long dynamics could encode an evolving DV, which is predicted from the CPP account emerging from classical studies in stationary environments, had not been tested. More specifically, signal processing choices in previous work (i.e., the application of a 1 Hz high-pass filter or decorrelation of the signal between consecutive samples in refs. 20,21 would have removed any sustained, across-sample activity. Thus, our use of a low high-pass filter cutoff (0.1 Hz) for the main analysis (Fig. 3) was critical for this aim in order to allow any slowly changing activity to evolve over the long trials in our task. We did observe ramping positivities over the duration of full trials at some central and frontal scalp locations (see Fig. 3A and Supplementary Fig. S4). However, using a regression analysis analogous to the one we used for MBL, we found that the amplitude of these signals did not

reflect evolving beliefs. Uncovering the functional role of these slow positive signals is beyond the scope of this study, but we note that their topographic distribution may be consistent with the previously described frontal encoding of elapsed time in non-human primates[24].

Our second research question related to how fast, sample-evoked centroparietal potentials encode momentary evidence. Centroparietal signals have previously been shown to scale with momentary evidence in expanded judgement, token-based tasks[20,21] in static environments where the optimal strategy is to perfectly accumulate the objective evidence in each token. The volatility in our task allowed us to go a step further and test whether centroparietal signals evoked by samples reflect objective evidence (here quantified as each sample's $|LLR|$), or rather effective evidence – that is, the belief update each piece of evidence imposes on a decision variable ($|\Delta\Psi| = |\Psi_{s+1} - \Psi_s|$). In static contexts, the normative accumulation strategy prescribes that objective and effective evidence be identical: if the evidence is accumulated perfectly, the step-change in a decision variable should be identical to the strength of the incoming objective evidence. In contrast, the non-linear transformation required by the normative model[23] in volatile contexts entails that changes in a decision variable depend not only on the objective evidence provided by each sample but also on its (in)consistency with the existing belief and the uncertainty under which it is encountered[16]. Our results show that centroparietal signals are driven by a sample's effective evidence after taking these factors into account according to a normative integration strategy (see Fig. 3E). This suggests centroparietal signals do not merely reflect objective sensory information, but rather an appropriate update quantity given the current belief.

It is worth noting that our results show strong encoding of belief updates in a broad group of electrodes, encompassing conventional CPP topographies but also extending to more anterior regions. However, the difference topographies comparing two simple models assuming sole encoding of objective vs. effective evidence (Fig. 3D and Supplementary Fig. 7) show that activity in posterior regions typically associated with the CPP is best captured by an effective evidence model, whereas the signal at anterior regions is equally well-explained by objective evidence (Supplementary Fig. 7A). Objective evidence in our task is inversely correlated with overall stimulus probability (i.e., samples with high $|LLR|$ values are generally less likely to occur over the task), which has long been associated with the frontal ERP component known as the P3a, most often linked with the detection of novel, but not necessarily decision-relevant, information[25,26]. Taken together, these results suggest that it is the posterior signal component that primarily handles the encoding of appropriate update quantities to be integrated in the decision process, but a frontal process may encode more general stimulus information content in parallel, here encompassing the objective evidence related to stimulus location in the current task.

Finally, we sought to directly link centroparietal signals to motor beta lateralisation and behaviour, exploiting variability in single-trial fluctuations[20,21]. We show that samples which evoked higher-than-expected ERP responses also had a greater weight on choice and that, in turn, motor fluctuations towards the end of the trial bias choice probabilities, in agreement with previous work[20,21]. We also observed that single-trial fluctuations in centroparietal responses modulated motor lateralisation updates, which additionally suggests that (1) weight modulations are implemented by a direct impact of single-trial fluctuations in centroparietal signals on motor beta, and (2) that single-trial variability in motor lateralisation can be at least partially accounted for by single-trial variability in centroparietal signals.

The above findings have implications for how computations resembling those prescribed by the normative model for our task[23] are implemented in the brain. The model specifies a two-step, sequential belief updating process wherein objective evidence derived from a new sample ($LLR_s$) is first combined with the existing prior belief ($\Psi_s$)

to yield a posterior ($L_s$), and only then is the posterior passed through the hazard rate-dependent non-linearity to yield the prior for the next sample ($\Psi_{s+1}$). Distinct representations of posterior and prior in this way can be beneficial because these variables should inform judgements about current states versus prediction of future ones, respectively[23]. However, this two-step process in the normative model contrasts with our finding that the CPP encodes an intermediary effective evidence quantity ($|\Delta\Psi_s|$) that itself captures the full effect of the non-linear transformation and is presumably fed forward for (perfect) accumulation in the evolving decision variable represented downstream in MBL. Effectively, our main findings appear to localise the critical normative transformation to the evidence to be accumulated, rather than the decision variable. Such a scheme may be advantageous from the standpoint of neural implementation as it would permit a single, fixed accumulation circuit to function equally well across environments with different levels of volatility. Indeed, we previously found that a neural circuit model originally developed to explain two-alternative decision-making in stable environments[27] could closely approximate normative belief updating on our task, but only following adjustment of a structural circuit parameter (the strength of recurrent excitation[16]) that would change the behaviour produced by the circuit in other task settings. Rescaling of evidence in the way we describe here would circumvent the need for adaptation of the decision circuit and thus may be an effective solution for producing flexible, adaptive decision-making in the brain.

Our study leaves some open questions. First, our findings suggest that in our volatile expanded judgement context, the CPP reflects the magnitude of individual belief updates evoked by single samples rather than a total DV integrated across samples. This integrated DV is instead encoded in the motor system. However, recent work has suggested that multiple brain areas may engage differently in decision formation depending on contextual factors such as knowledge about stimulus-response mapping, decision policy or integration strategy[8]. In our task, the stimulus-response mapping was known and constant throughout the task, so participants could directly forward appropriately transformed stimulus information into response preparation areas. However, the involvement of the CPP generators in decision formation & maintenance may be different in other task contexts. Indeed, our results beg the question of which signal might encode a decision variable in the context of a discrete token task where mappings are unknown. While in conventional continuous evidence tasks, the CPP (MBL) is compatible with the encoding of an unsigned (signed) DV, it is unclear what signal might maintain a sustained representation of a DV in a discrete token task like ours if neither the CPP nor MBL take on that role. Further research is required to better understand if and when the CPP and MBL interactions observed here generalise to other conditions.

Further, our finding that the CPP reflects the normatively required belief update means that the activity in its source regions must receive input beyond just signals carrying the newly arrived sensory evidence. However, there are several ways in which the non-linear transformation from objective to effective evidence could be implemented in the brain. Because belief updates in the normative model are determined by an interaction of incoming evidence with the state of the existing DV, one option is that the additional input to CPP generators (on top of objective sensory evidence) could take the form of feedback directly from areas representing the DV - in the case of the current task, putatively motor preparatory areas. In this case, local computations would need to take place to translate this combination of new evidence and current DV inputs into a signal coding for effective belief updates. An alternative possibility, inspired by our decomposition of belief updating into modulations of sensory evidence weighting by surprise and (at least in some contexts) uncertainty[16], is that the CPP generators receive inputs that relay internal estimates of these quantities, and these inputs, in turn, serve to modulate the representation of incoming

evidence. Work in predictive inference tasks with strong similarities to our own points to candidate cortical networks for representing surprise and uncertainty[28,29], while our own previous work with the current task also implicates pupil-linked arousal systems[16]. A final plausible option is that evolving beliefs may bias early sensory responses, directly affecting the primary input of the CPP generators without the need for any additional mechanism. Previous work on simple decision tasks has shown that beliefs and expectations can bias early sensory representations of evidence[30], or influence the accumulation process itself[31]. In our task, a sample's location on the screen indicated how much evidence it provided. However, in the present case we cannot assess the strength of sensory representations independently from the evidentiary quantity they are associated with. Thus, testing these or other plausible candidate explanations for the implementation of effective evidence encoding will be a matter to be addressed in future work.

In sum, this study sheds new light on the neural correlates of normative belief updating. We show that centroparietal signals in this task context encode effective evidence provided by each sample – that is, the impact of new information on beliefs - rather than tracking the evolving DV across samples. Our results expand our understanding of which precise aspect of decision formation is tracked by the process reflected in the CPP and further extends its validity as a marker of decision processing to volatile contexts, which require non-linear integration and, to the best of our knowledge, had not been examined before. This broadens our understanding of the nature of centroparietal decision signals and paves the way for future work studying the link between the different neural signals involved in decision-making.

## Methods

### Participants
Twenty participants aged 19–35 ($M = 23.1$, SD = 3.55; 15 male) took part in two EEG sessions while performing a decision-making task (Fig. 1A). All procedures were approved by the human research ethics committee of University College Dublin. All participants provided written informed consent and were compensated for their time (10€/hour).

### Behavioural task
Stimuli were coded using Psychtoolbox-3[32] for Matlab (Mathworks). Participants monitored a set of checkerboard patches ('samples') appearing anywhere along a semicircular arc in the lower visual hemifield and centred on fixation. Sample locations were drawn from one of two overlapping Gaussian distributions, with equal s.d. and means left vs right of the vertical meridian (mean polar angle relative to vertical meridian: +/− 26°, s.d.: 29°). Samples were presented for 300 ms, and a blank 100 ms interval separated consecutive stimuli. Due to a small stimulus timing discrepancy during recording, these timings were 7 ms shorter (293 ms/93 ms) in 7 participants. All our analyses account for this. For whole-trial grand-average ERP traces, data were aligned simply by appending 14 ms worth of NaN at the end of each token period in those participants. Participants viewed a maximum of 10 samples per trial, and trials had a 0.3 probability of terminating early, in which case the number of samples was drawn from a uniform distribution between 1–9. The distribution from which sample locations were drawn could change during the trial with a fixed probability (hazard rate, H) of 0.1. Participants' task was to report what they believed to be the 'active' distribution at the end of each trial (left- or right-centred).

Stimuli were presented against a grey background. Three placeholders were present throughout each trial: a light-grey vertical line extending downward from a fixation; a light-grey half-ring in the lower visual hemifield (polar angle, from − 90 to + 90°; constant eccentricity of 8.1°) and a fixation mark which changed colours to indicate trial time (blue: intertrial period and initial fixation, red: view samples, green:

response cue). The position of samples along the grey half-ring indicated their LLR: samples appearing close to the vertical meridian (0°) provided weak evidence (i.e., small |LLR|), as they could be drawn from either the left- or right-centred distribution with similar probabilities. Instead, samples appearing close to the horizontal midline (− 90/+ 90°) provided strong evidence (i.e., large |LLR|), as they were more likely to be drawn from one of the two distributions than the other. All trials started with a checkerboard patch presented at a polar angle of 0° (i.e., on the vertical midline) for 300 ms, serving as a warning cue, followed by the first sample and a change in the colour of the fixation point (red) after a 100 ms blank interval. Evidence samples consisted of black and white flickering checkerboards (temporal frequency, 10 Hz; spatial frequency, 2°) within a circular aperture and varied in polar angle. At the end of the trial, the fixation point changed to green ('Response cue') 0.7 s to 1.3 s (uniform distribution) after the final sample offset. Participants reported their response using their left-or right-hand thumb to press the left or right arrow keys, respectively, on a standard keyboard. There was no response deadline and no time pressure to respond. Auditory feedback 0.1 s after the response informed participants of their accuracy. An ascending tone (350 Hz to 950 Hz) indicated correct responses, and a descending one (950 Hz to 350 Hz) errors. The next trial started after a 2 s intertrial interval.

### Behavioural analyses
Normative belief updating requires the following computation:

$$L_s = L_{s-1} + LLR_s \tag{1}$$

$$\Psi_s = L_{s-1} + \log\left(\frac{1-H}{H} + \exp(-L_{s-1})\right) - \log\left(\frac{1-H}{H} + \exp(L_{s-1})\right) \tag{2}$$

Here, $L_s$ is the observer belief after encountering the evidence in sample s, expressed in log-posterior odds of the alternative task states; H is the hazard rate; $\Psi_s$ is the prior expectation of the observer before encountering a sample at location $X_s$, and $LLR_s$ is the log-likelihood ratio:

$$LLR_s = \log\left(\frac{p(X_s|Right)}{p(X_s|Left)}\right) \tag{3}$$

where $p(X_s|Right)$ and $p(X_s|Left)$ refer to the conditional probability of the token appearing at the observed location X for the current sample, given the generative distribution at that time is centred on the right and left, respectively.

We used this model to compute a metric of surprise and a metric of uncertainty[16]. Surprise was quantified as the analytically computed change-point probability after each sample. Note that we adopt the term "surprise" here in place of change-point probability[16] to avoid ambiguity in acronym use; CPP, in this paper, refers to centroparietal positivity. The change-point probability indicates the probability that a change of state has just occurred based on the most recent sample's likelihood under both generative states, the belief prior to that sample, and the generative H:

$$Surprise_s = \left(\frac{(H * ((p(X_s|Left) * p(Left_{s-1})) + (p(X_{s-1}|Right) * p(Right_{s-1}))))}{\begin{array}{c}(H * ((p(X_s|Left) * p(Left_{s-1})) + (p(X_s|Right) * p(Right_{s-1})))) \\ + (1-H) * ((p(X_s|Left) * p(Left_{s-1})) + (p(X_s|Right) * p(Right_{s-1})))\end{array}}\right) \tag{4}$$

where p(Left) and p(Right) represent the probability that the generative state is left or right based on the prior belief and are computed

as follows:

$$p(Left_{s-1}) = \frac{1}{e^{L_{s-1}}+1} \quad (5)$$

$$p(Right_{s-1}) = \frac{e^{L_{s-1}}}{e^{L_{s-1}}+1} \quad (6)$$

Surprise values were logit transformed in order to reduce skewness prior to introducing them in the behavioural and EEG regressions.

Uncertainty was defined as the negative absolute prior ($-|\Psi|$). Higher values of uncertainty thus correspond to weak prior beliefs. Surprise and uncertainty are not explicitly used in the normative computations, but provide an intuitive understanding of the impact the non-linear transform has on belief updates and have been used in previous research[16]. It is worth noting that as environmental volatility (i.e., H) increases, the normative nonlinearity becomes steeper, producing faster transitions between strong beliefs and reducing the amount of time spent in high uncertainty states. A similar effect will also be achieved by increasing the signal-to-noise ratio (SNR; the difference in means divided by s.d.) of the generative distributions. The generative volatility ($H = 0.1$) and SNR (1.79) were both higher than those in previous work (e.g., $H = 0.08$ and $SNR = 1.17$ in ref. [16]). Thus, given the lower proportion of samples expected to be received in high uncertainty states with the current generative settings, the modulation by this factor is expected to be weaker in our task.

## Weight on choice

We quantified the impact of evidence at each sample position on observed or model-derived choices, and the modulation of this impact by surprise, via logistic regression:

$$\begin{aligned}Logit(P(choice_{trl} = right)) \sim \beta_0 + \sum_{i=1}^{10} \beta_{1,i} LLR_{i,trl} \\ + \sum_{j=2}^{10} \left( \begin{array}{l} \beta_{2,j} LLR_{j,trl} * Surprise_{j,trl} + \\ \beta_{3,j} LLR_{j,trl} * Uncertainty_{j,trl} \end{array} \right) \end{aligned}$$
$$(7)$$

where i and j index sample position within sequences of 10 samples, and LLR was the true LLR. We restricted this analysis to full trials (i.e., trials where the maximum possible number of samples (10) were presented).

## Computational modelling

We fit the normative model to participants choices, along with several alternative models that capture some, but not all, of its features: a perfect accumulation model, a perfect accumulation model with non-absorbing bounds, and a leaky accumulator. All models included noise and evidence gain (i.e., linear multiplicative scaling of LLR) parameters. The noise term was used to corrupt the log-posterior odds at the end of each trial, and choice probabilities in all models were computed as $0.5 + 0.5*erf(L/noise)$.

In addition, subjective hazard rate, bound and leak parameters were estimated for the normative, non-absorbing bounds and leaky accumulator models, respectively. We fitted the model parameters by minimising the cross-entropy between participant and model choices. Error was minimised via particle swarm optimisation (PSO), setting wide bounds on all parameters and running 300 pseudorandomly initialised particles for a maximum of 1500 search iterations.

The goodness of fit of the investigated models was assessed by computing Bayes Information Criterion (BIC):

$$BIC = 2 * e + k * \log(n) \quad (8)$$

where k is the number of free parameters, n the number of trials and e the model's error as measured by the cross-entropy value.

## Data acquisition and analysis

**Preprocessing.** 128-channel scalp EEG data were recorded with a sampling rate of 512 Hz (Biosemi). Simultaneously, eye gaze and pupil size were recorded using the Eyelink Plus 1000 Tower system (SR Research, ON, Canada) recording at 500 Hz. EEG data were processed using Matlab R2021a (MathWorks) and the EEGlab v-2021.0 toolbox[33] and ERPlab v-8.10 plugin[34]. EEG data were high-pass filtered above 0.1 Hz (3rd order, two-pass Butterworth), and notch filtered at 50 Hz and 100 Hz to remove line noise. Data were epoched from [−0.4 s to 5.4 s] around the trial start and were baselined using the 100 ms prior to the trial onset cue. Eye movement artifacts were identified and removed using EEGlab's runica function, which applies the extended Infomax algorithm for Independent Component Analysis (ICA)[35]. In addition, channels showing ±150 μV deviations from baseline were interpolated on a trial-by-trial basis using the TBT toolbox[36]. If any given epoch had more than 20 bad channels, it was marked for rejection. Further, if any given channel was marked as artifactual on >30% of epochs, it was interpolated across the whole experiment. Data were downsampled to 200 Hz, and finally, we applied a Current Source Density (CSD) filter using the CSD toolbox v-1.1[37] to the preprocessed data to reduce the spread of potential components on the scalp through computing a second spatial derivative.

**Spectral analysis.** We used a sliding-window Fourier transform to compute time-frequency decompositions of the single-trial activity. We used a Hanning taper (window length, 0.4 s; time steps, 0.005 s; frequency steps, 1 Hz) for the range 1–35 Hz using the FieldTrip toolbox[38]. To compute motor lateralisation, we averaged a cluster of 6 left- and 6 right-hemisphere electrodes based on the response-aligned topography differences between left- and right-response trials, and computed an index of motor lateralisation as the difference in beta power (13–30 Hz) between the two hemispheres (left minus right, electrodes highlighted in figures). Time-frequency data were decibel (dB) transformed prior to analysis (10 x log10 (power/baseline)), consistent with[16], where the baseline was the trial-averaged power in the 100 ms preceding trial onset).

## EEG modelling analysis

Following Murphy et al.[16], we ran a time-resolved regression to investigate whether motor beta lateralisation encoded computational variables relevant for normative belief updating:

$$MBL_{s,t} \sim \beta_0 + \beta_1 \Psi_s + \beta_2 LLR_s + \beta_3 LLR_s * Surprise_s + \beta_4 session \quad (9)$$

The modulation, by surprise, is predicted by the non-linear prior scaling in the normative model and was observed in the behavioural data, but they are not explicitly used in the computations. Thus, to directly test whether changes in motor lateralisation following each sample reflected the normatively required belief update, we ran a second simplified regression using only the prior belief ($\Psi_s$) and the belief update ($\Psi_{s+1} - \Psi_s$) as regressors:

$$MBL_{s,t} \sim \beta_0 + \beta_1 \Psi_s + \beta_2 (\Psi_{s+1} - \Psi_s) + \beta_3 session \quad (10)$$

Next, we ran an equivalent analysis for the ERP to test whether centroparietal activity reflected evidence-related variables relevant to normative belief updating. (Eqs. 11,12). To reduce noise in the estimated coefficients, EEG signals were low-pass filtered at 6 Hz (two-pass

Butterworth filter) prior to the analysis:

$$EEG_{e,s,trl} \sim \beta_0 + \beta_1 |\Psi|_s + \beta_2 |LLR|_s + \beta_3 |LLR|_s * Surprise_s + \beta_4 session \quad (11)$$

$$EEG_{e,s,trl} \sim \beta_0 + \beta_1 |\Psi_{s+1} - \Psi_s| + \beta_2 session \quad (12)$$

In all regressions (Eqs. 9–12), the analysis was restricted to full trials (i.e., trials where the maximum possible number of samples (10) were presented), and a session covariate taking values 0 or 1 for sessions 1 and 2, respectively was included to control for spurious differences between the two recording sessions. Analyses were run for each sample separately, for a period spanning 1 s after sample onset, and finally, coefficients were averaged across samples. Note that for broadband EEG regressions (Eqs. 11–12) absolute values for the $\Psi$ and LLR regressors are used. This is because previous research has shown that centroparietal potentials evoked by decision-relevant evidence are always positive, irrespective of the response alternative supported by the evidence[11], consistent with a scheme where scalp amplitude reflects the absolute value of what is internally encoded as a signed decision variable[39]. Further, the initial sample was excluded from analysis in Eq. 9 and Eq. 11 because prior ($\Psi$) values are always 0 at the beginning of the trial.

## EEG model comparison

To test the extent to which a specific computational quantity explained the EEG variability better than an alternative one, we compared the adjusted R-squared values of simple regressions, including only the computational variable of interest to a reference model. In the main text, we compared models reflecting three possible aspects of a belief updating process that could be captured by the CPP and MBL: a normative decision variable ($\Psi$), momentary objective evidence (OE, quantified as the LLR), or effective evidence (EE, quantified as $\Delta\Psi$ and indicating the extent to which a belief is updated following any given sample. Absolute and signed quantities were used for the CPP and MBL analysis, respectively. In an additional supplementary analysis for the CPP (Supplementary Fig. S7), we further compared these models to an intercept-only model and to a model containing only surprise to investigate whether sensitivity to surprise alone could account for the observed results. Only trials where the maximum possible number of samples (10) were presented were included in all analyses. Further, in order to increase our signal to noise ratio, and since the key comparisons of interest pertained to the sample-wise evoked potentials, all these simple regressions were run after additionally high-pass filtering the preprocessed data above 1 Hz (3rd order, two-pass Butterworth; see next section).

## Residuals analysis

Finally, we harnessed the single-trial variability of EEG signals to investigate whether intrinsic fluctuations in neural activity over and above the effects captured by the normative model influenced behaviour. We used the residuals from Eq. 9 and Eq. 12 as measures of trial-by-trial fluctuations not accounted for by the normative model.

For the analysis of residual variability in centroparietal signals, we tested whether samples evoking particularly big positivities (i.e., positive residuals) were given more weight on choices by including the residuals of Eq. 12 as an interaction term with LLR on a regression

predicting choice (Eq. 13):

$$\text{Logit}(P(choice_{trl} = right)) \sim \beta_0 + \sum_{i=1}^{10} \begin{pmatrix} \beta_{1,i} LLR_{i,trl} + \\ \beta_{2,i} LLR_{i,trl} * Residual\ CPP_{i,trl} \end{pmatrix} + \sum_{j=2}^{10} \begin{pmatrix} \beta_{3,j} LLR_{j,trl} * Surprise_{j,trl} + \\ \beta_{4,j} LLR_{j,trl} * Uncertainty_{j,trl} \end{pmatrix} \quad (13)$$

where i and j index sample position within sequences of 10 samples. Only trials where the maximum possible number of samples (10) were presented were included in this analysis. Note that, in order to increase our signal-to-noise ratio for this single-trial analysis of centroparietal signal residuals, we repeated the analysis in Eq. 12 after additionally high-pass filtering the preprocessed data above 1 Hz (3rd order, two-pass Butterworth). This method has been used for similar residuals analysis in the past[20], and it overcomes two problems. First, it avoids the need to baseline-correct data prior to each individual sample within a trial, which might be problematic given the timings of our stimuli and overlapping potentials. Second, since our analysis involves averaging over samples occurring at various times after trial onset (ranging between 0 and 4.4 s), removing slow drifts in the data, which are not of interest in this analysis, makes activity following early and late samples more comparable and increases the signal to noise ratio. Importantly, the effects of interest did not change qualitatively or quantitatively when repeating the regression analysis on this additionally filtered dataset (see Supplementary Fig. S3). Residuals were extracted from the analysis reported in Supplementary Fig. 3B from a subset of anterior and posterior channels centred around the midline, where effects were found to be maximal. Note that the 1 Hz high-pass filter settings was exclusively used for this residuals analysis and additional model comparison results, not for the main analyses of CPP activity reported in Fig. 3B, C.

Regarding the impact of fluctuations in beta lateralisation on choices, we investigated whether variations in lateralisation following any given sample modulated choice probabilities by including the residuals from Eq. 9 as a main effect (Eq. 14).

$$\text{Logit}(P(choice_{trl} = right)) \sim \beta_0 + \sum_{i=1}^{10} \beta_{1,i} LLR_{i,trl} + \sum_{j=2}^{10} \begin{pmatrix} \beta_{2,j} LLR_{j,trl} * Surprise_{j,trl} + \\ \beta_{3,j} LLR_{j,trl} * Uncertainty_{j,trl} + \\ \beta_{4,j} Residual\ lateralisation_{j,trl} \end{pmatrix} \quad (14)$$

Finally, we aimed to directly link fluctuations in centroparietal activity to changes in motor beta lateralisation following each sample. Motor beta lateralisation [left-right] was averaged over the first 100 ms following each sample, and we computed a measure of motor updates ($\Delta$MBL) indicating how much beta lateralisation changed between consecutive samples (Eq. 15), as well as a measure of the normatively required update, defined as the change in normative priors between consecutive samples (Eq. 16).

$$\Delta MBL_s = MBL_{s+1} - MBL_s \quad (15)$$

$$\Delta\Psi_s = \Psi_{s+1} - \Psi_s \quad (16)$$

Then, we tested whether variability in motor updates could be partially explained by fluctuations in centroparietal signals. In other words, whether centroparietal fluctuations modulated the extent to which motor updates deviated from the normatively required change.

To test this, we fitted the following regression (Eq. 17):

$$\Delta MBL_s = \beta_0 + \beta_1 \Delta\Psi_s + \beta_2 \Delta\Psi_s * ResidualCPP_{s,\,trl} \tag{17}$$

## Cluster statistics

EEG regression results were statistically evaluated using cluster-based permutations using the FieldTrip toolbox[38,40]. All tests were two-tailed, cluster-corrected ($p < 0.05$), and 10000 permutations were run. For analysis in Eq. 11 and Eq. 12, regression coefficients were averaged over a cluster of preselected centroparietal electrodes prior to testing. The preselected electrodes encompassed areas where the centroparietal positivity is typically observed in evidence accumulation tasks.

## Reporting summary

Further information on research design is available in the Nature Portfolio Reporting Summary linked to this article.

## Data availability

All raw EEG and behavioural data in this study have been deposited in the OSF database: https://doi.org/10.17605/OSF.IO/NHQEV.

## Code availability

All analysis code used to produce the data and figures presented in this paper is available at: https://doi.org/10.17605/OSF.IO/NHQEV.

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

## Acknowledgements

This work was supported by a grant from Science Foundation Ireland (now Taighde Éireann) under grant number 19/US/3599 and an Investigator Award from the Wellcome Trust under grant number 219572/Z/19/Z (to S.P.K), and the EU Horizon 2020 Research and Innovation Programme, Marie Skłodowska-Curie Grant Agreement No. 843158 (to P.R.M.). The authors would also like to thank students Liang Tong and Bowen Duan for their assistance during data collection.

## Author contributions

Conceptualisation: E.P.P., P.R.M. & S.P.K; software: E.P.-P. and P.R.M.; data curation & analysis: E.P.-P.; resources: S.P.K.; writing of original draft: E.P.-P.; review & editing: E.P.-P., S.P.K. & P.R.M.; supervision: S.P.K. & P.R.M.

## Competing interests

The authors declare no competing interests.
