## [Transparent Peer Review file · Nature Communications]

Dissociable encoding of evolving beliefs and momentary belief updates in distinct neural decision signals

Corresponding Author: Dr Elisabeth Pares-Pujolras

Version 0:

Reviewer comments:

Reviewer #1

(Remarks to the Author)

In this manuscript, the authors used electroencephalography (EEG) to search for human brain signals that monitor and integrate uncertain sensory information to make a perceptual decision. They contrasted two such signals, one over the centro-parietal electrodes known as CPP (centroparietal positivity) that appears to mainly monitor sensory information, and the other more lateralized known as MBL (motor beta lateralisation) that instead appears to accumulate evidence towards the decision. By looking at the fluctuations from trial to trial, they argue that CPP feeds into MBL, thereby providing a path from early sensory information intake to late motor behavior. The experiments and analyses are carefully carried out, but I have some reserve on the novelty and the strength of their conclusions.

First, this work follows the same experimental paradigm as another paper recently published by the senior author (Murphy et al., 2021, Nat Neuro). In particular, they use the same procedure to deliver a stimulus step by step (every 400 ms), with a hazard rate of about 10% to possibly change the distribution at the origin of the stimulus within a trial. The same model is used in the original paper and in the current manuscript (Equations 1 and 2 are identical, and so is the text around these equations). The theoretical question to find a path from sensory to decision is also the same. The main difference appears to be that the current manuscript uses EEG while the previous paper was using a different imaging technique (magnetoencephalography). In itself, I do not have a problem replicate an interesting experiment, but I have not found a motivation for why EEG was going to give us new insights. On the contrary, it looks like the authors are able to perform fewer analyses with EEG than they were able to do with MEG (all the frequency-based analyses make little sense with EEG). So, I am left searching for the groundbreaking novelty in the manuscript that would make it appeal to the readership of Nature Communications.

Second, I find that the main conclusions that CPP and MBL are performing different computations, and that one (CPP) feeds into the other (MBL), are not very convincing. On the first part, different analyses are performed for CPP and for MBL. This is blatantly obvious when one tries to compare Figures 2 and 3 that summarize the analyses for CPP and MBL respectively. In particular, Equations 9 and 10 are used in Figure 2, and Equations 11 and 12 are used in Figure 3. Another example is the use of a high-pass filter to analyze CPP (page 12, line 333) that as far as I understand, was not used for MBL. Finally, I do not understand how the authors can assert that CPP is not doing any integration when one looks at Figure 3A, where the EEG amplitude for the first few samples is clearly smaller than the last few.

On the second part of the conclusions, the evidence for CPP feeding into MBL is relatively small. It looks like that this conclusion comes mostly from the analysis shown in Figure 4F, but the effect seems to disappear when the analysis is split between early and late samples (right panel). Altogether, the conclusions are appealing, but the evidence is not very convincing.

Third, informed by the conclusions of the previous paper (Murphy et al., 2021), I was expecting that the authors would test new predictions. For instance, one could have tested different values of the hazard rate (including a baseline condition where this rate is zero), to check whether CPP would be sensitive to the hazard rate. One could have added a secondary task to ask participants if they detected a change of distribution within a trial, and analyze the EEG signals separately depending on whether participants did indeed detect a change. Again, I think the analyzes presented here are well performed, but the project lacks ambition.

(Remarks on code availability)

Reviewer #2

(Remarks to the Author)

In this study, Parés-Pujolràs and colleagues investigate the functional role of two important neural signatures found in evidence accumulation and decision-making. They employed an experimental paradigm designed to be able to disentangle their roles, in particular, with a lower time scale allowing them to study what happens at each sample of evidence delivered.

Using an elegant combination of computational modeling and electrophysiology, it was found that the centroparietal positivity signal relates to belief update sample by sample, rather than an overall decision variable evolution throughout the samples. Via a smart use of the “residuals analysis approach”, another key finding is linking the fluctuations in the centroparietal encoding of belief updates to the variability in motor beta lateralisation encoding of the decision variable.

Please note that my expertise lies primarily in decision-making and belief updating, rather than EEG.

The paper is particularly well written and very clear, because the authors describe their reasoning motivating each piece of analysis. Nevertheless, I have some comments:

General comments

1. Have the authors have tried to apply a 1 Hz high pass filter for the main analysis as in previous work rather than the 0.1 Hz low pass filter, to support their discussion point that it matters for identifying any sustained activity across the samples? (not only slow drifts but relevant activity?)
How does Fig S3 contribute to this point?
2. Have the authors analysed separately the trials that actually follow a true change point (not only the change points occurring during the sequence of samples)? Or is it not relevant because we cannot pinpoint exactly at which sample/which trial the participant picks up on the change point that occurred? (I would have expected the effect of effective evidence to be stronger after a true change point as compared to all sampled pooled)
3. The residual effect for motor preparation did not vary between early and late samples, unlike the MBL effect; however, there seems to be a small difference later after sample presentation (Fig 4F), do the authors think it could be a meaningful late effect?
4. Do the authors think that the CPP results would generalise to tasks where mappings between stimulus and response are not known beforehand, or can change before the response is provided?
5. Even though the Results section describe which findings replicate the MEG results of Murphy et al., it would be interesting to have a broader discussion of how much overlap/whether any differences, also, are seen, in the Discussion section or Supplement
6. It would be useful that the introduction and results section better describe the non linearity aspect between the posterior belief and the prior of the next sample for readers unfamiliar with the volatility model of Glaze et al.

Minor comments

7. The notion of stationary stimulus: RDK may not be characterised as stationary but indeed too fast as the authors argue; perhaps another word that ‘stationary’ would fit better?
8. Could the authors specify participants’ compensation (nature and amount)
9. In the introduction, could the authors develop the biases they refer to in “CPP subject to additional task-dependent biases (line 100)”

(Remarks on code availability)

Reviewer #3

(Remarks to the Author)

Pares-Pujolras and colleagues examine electrophysiological candidates for a decision variable in a complex task in which

momentary evidence must be updated dynamically in accordance with the possibility that the correct response can change over time. The authors show that motor beta lateralization evolves slowly and matched the computational profile of a normative decision variable, whereas parietal event triggered responses (centroparietal positivity) better matched signals for moment to moment belief adjustments, rather than an accumulated evidence signal. The authors show that both signals relate to behavior in ways that support these computational categorizations and furthermore, that residual fluctuations in CPP strength covary with the degree to which new observations lead to changes in beta lateralization.

Overall, I found this paper to be interesting, timely, and well communicated. Nonetheless, I have a few questions and concerns about the manuscript in its current form.

One question I have is whether participants make horizontal saccades when seeing the most disconfirmatory information, and whether any such eye-movements might contaminate the overarching results. It is clear that successive stimuli that alternate between left and right sides of the screen would drive large belief updates in the model and likely be interpreted as changepoints. But I can't tell from the methods how big the arc is, and to what extent participants might break fixation to gaze at new stimuli rather than just viewing them in periphery. CPP doesn't seem like the sort of ERP that would be elicited by eye movements, but given that some of the key variables almost certainly relate to alternative lateral screen positions, I think it is worth considering eye movements carefully.

How distinct are $|LLR|$ and $|LLR|^*$ surprise? I would think that these things are pretty highly correlated, and only distinguished by some pretty specific task situations... It sort of seems like what the authors are trying to get at with the latter regressor is the idea that the response to a given evidence will be stronger if that evidence is on the opposite side of the screen as the decision variable up to that point in time. As a complement to the regression analysis, I could imagine just looking at this directly, but plotting the magnitude of CPP as a function of evidence location separately for cases where preceding decision variable was on either same or opposite side of screen.

While I definitely understand the a-priori focus on the CPP and beta lateralization, I was left wondering whether these signals are unique, or whether there might be other similar signals that would emerge if the same regression analysis were examined across the sensor space.

If I understand the model correctly, the sigmoid is essentially modeling the transition function of the task, which when applied in the space of log probability ratios, leads to floor and ceiling corresponding to the minimum probability for either state (due to the hazard rate). Is that correct? And if so, are the differences in updating from trial to trial mainly due to whether the evidence pushes the DV toward an extreme (in which case it gets pushed back by the transition function) versus towards complete uncertainty (in which case it is totally unaffected by the transition function)? Assuming my understanding is correct, I guess I wonder whether these processes are expected to happen at the same time... because it seems plausible to me that one might update their decision variable with respect to the evidence, and hold off on applying the transition function, perhaps until observing the next piece of evidence. Basically, my point here is just that I think that the authors are making some assumptions about when different stages of computation are happening with their analyses, and I think it would be really good for them to make those assumptions clear and justify them where possible.

Minor:

Topoplots in 3B could use a bit more labeling, I figured out color scheme eventually, but it was not obvious to me.

(Remarks on code availability)

Version 1:

Reviewer comments:

Reviewer #1

(Remarks to the Author)

I would like to thank the authors for taking my comments, and those of the other reviewers, very seriously. I now understand better the novel contributions of the current manuscript. Regarding the issue of the slow positive drift that is visible in Figure 3A (last part of my second point), I find the response of the authors (top of page 6 of the reply to the reviewers) quite interesting, and I would encourage the authors to include a summary of this response in the main text (probably around line 329).

(Remarks on code availability)

Reviewer #2

(Remarks to the Author)

In their revision, the authors have now addressed the points raised earlier.

A minor final point would be to clarify the reasoning underlying the motivation for using EEG, as there is a degree of circularity in the reasoning: the authors say they need EEG to get the CPP as revealing intermediate steps of evidence accumulation; but also, that their findings help further characterise the nature of the CPP signal (specifically, as belief updating).

(Remarks on code availability)

Reviewer #3

(Remarks to the Author)

The authors have fully addressed my concerns.

(Remarks on code availability)

REVIEWER COMMENTS

Reviewer #1 (Remarks to the Author):

In this manuscript, the authors used electroencephalography (EEG) to search for human brain signals that monitor and integrate uncertain sensory information to make a perceptual decision. They contrasted two such signals, one over the centro-parietal electrodes known as CPP (centroparietal positivity) that appears to mainly monitor sensory information, and the other more lateralized known as MBL (motor beta lateralisation) that instead appears to accumulate evidence towards the decision. By looking at the fluctuations from trial to trial, they argue that CPP feeds into MBL, thereby providing a path from early sensory information intake to late motor behavior. The experiments and analyses are carefully carried out, but I have some reserve on the novelty and the strength of their conclusions.

First, this work follows the same experimental paradigm as another paper recently published by the senior author (Murphy et al., 2021, Nat Neuro). In particular, they use the same procedure to deliver a stimulus step by step (every 400 ms), with a hazard rate of about 10% to possibly change the distribution at the origin of the stimulus within a trial. The same model is used in the original paper and in the current manuscript (Equations 1 and 2 are identical, and so is the text around these equations). The theoretical question to find a path from sensory to decision is also the same. The main difference appears to be that the current manuscript uses EEG while the previous paper was using a different imaging technique (magnetoencephalography). In itself, I do not have a problem replicate an interesting experiment, but I have not found a motivation for why EEG was going to give us new insights. On the contrary, it looks like the authors are able to perform fewer analyses with EEG than they were able to do with MEG (all the frequency-based analyses make little sense with EEG). So, I am left searching for the groundbreaking novelty in the manuscript that would make it appeal to the readership of Nature Communications.

> We'd like to thank the reviewer for their comments, which have helped us to clarify the key novel aspects of our work.

Using MEG, Murphy et al. (2021) was able to trace the encoding of objective sensory evidence in each token through visual cortical areas, and at the far end of the sensory-motor pathway, a motor plan reflecting the evolution of the normative decision variable across tokens. The intermediate neural processing steps required to transform the former to the latter, however, were not identified. In particular, the study identified no neural correlate of the critical intermediary "belief update" computation, that nonlinearly transforms the objective evidence carried by each token in order to dampen evidence that agrees with an already held belief. This is a core, defining quantity in the normative solution to decision making under volatile conditions, and as such represented a vital missing link in the previous work. The reason it was critical to employ the same paradigm and computational framework with EEG is that EEG held significant promise as a means by which to probe those intermediate processing levels. Specifically, the centroparietal positivity (CPP) is well-established to reflect an intermediate cognitive processing stage that is neither sensory nor motor in character, and that has been found to encode an abstract (effector-independent) representation of cumulative evidence in standard non-volatile tasks with continuous stimuli. Since no equivalent intermediary signal has been conclusively identified in MEG, using EEG was essential to our aims and the CPP was our key neural signal of interest.

In addition to identifying a neural correlate of this intermediary quantity for the first time, our findings also shed new light on the CPP itself as a neural correlate of evidence accumulation. Up to now, a substantial number of studies had found that in perceptual judgments of

continuous stimuli such as random dot motion or contrast discrimination, the CPP exhibits all of the characteristics of a decision variable (DV) that linearly and continuously accumulates evidence from start to finish (e.g. O'Connell et al. 2012; Kelly & O'Connell 2013). The task we used here differs from most previous EEG evidence accumulation work in two key respects: 1) the use of discrete tokens, and 2) the volatility of the environment. Given the extant CPP literature, the most obvious predictions would have been that the CPP reflects either the token-by-token decisions about the momentary objective stimulus information (as previously suggested - Wyart et al., 2012, 2015), or the state of a decision variable evolving across tokens. Here we find that the CPP reflects neither of these things in this context, but rather the belief update quantity that had been missing from the previous MEG work on this paradigm. This constitutes a major advance in our understanding of the CPP as a key neural marker of decision formation, and highlights the potentially key role of its neural generators in affording behavioural flexibility across different decision-making contexts.

Second, I find that the main conclusions that CPP and MBL are performing different computations, and that one (CPP) feeds into the other (MBL), are not very convincing. On the first part, different analyses are performed for CPP and for MBL. This is blatantly obvious when one tries to compare Figures 2 and 3 that summarize the analyses for CPP and MBL respectively. In particular, Equations 9 and 10 are used in Figure 2, and Equations 11 and 12 are used in Figure 3.

As stated in the paper (p.30, lines 857 onwards), it was necessary to adapt our regression equations to the known generative properties of the two signals.

The CPP regressions must take absolute values of the Ψ (prior belief) and LLR regressors because the CPP is known to always take a positive sign, regardless of the choice supported by the evidence, and to scale with the absolute strength of evidence (Kelly & O'Connell 2015). In contrast, the sign of beta lateralisation does depend on the movement supported by the evidence, and so we therefore used signed values of Ψ and LLR to interrogate this signal. To illustrate the importance of using signed regressors for signed signals and unsigned regressors for unsigned signals, below we show that re-running the analysis using signed regressor values for CPP (Fig. R1.A) and absolute regressor values for MBL (Fig. R1.B) produces null results in both cases. Thus, ignoring the signed vs unsigned nature of these two signals leads to a failure to capture the key sensitivities of either signal, highlighting that it was necessary to adjust the regression equations to allow for these signal differences.

A second difference is that, whereas Eq. 10 for MBL analysis includes a Ψ regressor, Eq. 12 for the corresponding CPP analysis does not. This is because in the case of the CPP we found no significant encoding of $|\Psi|$ in the previous analysis, and therefore excluded it from subsequent analysis. To reassure the reviewer that this does not affect our conclusions, we have re-run the analysis in Figure 3C including the $|\Psi|$ regressor (see Fig. R1C below) to show that its inclusion does not change the results.

We thank the reviewer for sharing this perspective and thereby prompting us to ensure that the adaptation of regression equations to the two signals is clearly justified and well-suited to test our hypotheses.

Fig. R1. A. Regression coefficients for an alternative version of Eq 12 (Eq. S1 shown at inset) where signed instead of absolute values of $\Delta\Psi$ were included as a regressor of CPP activity. No significant clusters were identified and the effect topography did not have a centro-parietal focus as expected of a CPP component. This is expected because of a basic signal property of the CPP, whereby it builds with positive sign regardless of which alternative is favoured. **B.** Regression coefficients for an alternative version of Eq 10 (Eq. S2) where absolute instead of signed values of Ψ (prior belief) and $\Delta\Psi$ (belief update) were included as regressors of MBL activity. No significant clusters were identified. This is similarly expected because MBL is a signed signal, going positive or negative depending on the favoured alternative. **C.** Regression coefficients for an extended version of Eq. 12 (Eq. S3) including both $|\Psi|$ and $|\Delta\Psi|$ as regressors. As in the main paper, no significant encoding of prior emerged. Topographies illustrate the standardised regression coefficients of the $|\Delta\Psi|$ effect at the shaded time. *Asterisks along the bottom indicate significant cluster periods for tests averaging over the highlighted electrodes. Y-axes are the same as in the corresponding figures in the main paper.

Another example is the use of a high-pass filter to analyze CPP (page 12, line 333) that as far as I understand, was not used for MBL.

> Again, this reflects a difference in basic signal properties. The CPP is a broadband ERP signal that is vulnerable to slow drift, and to remove any such slow artifacts we high-pass filtered the EEG data with a 0.1 Hz cutoff which removes cycle lengths slower than 10 sec while keeping any relevant signals evolving on the timescale of the experimental trial (4 sec), such as an evolving prior if it had been truly present. The MBL signals were in fact derived from the same preprocessed EEG, including the 0.1 Hz high pass filter, but after time-frequency decomposition, spectral amplitude waveforms are not vulnerable to slow drift and thus do not require additional high-pass filtering; in fact, it is best not to do so because, to avoid filter edge artifacts, the 0.1 Hz filtering would have to be applied to beta-band power computed over raw continuous data from the whole block, which in turn precludes important artifact-removal processing steps applied to trial-segmented data that improve signal-to-noise ratio (e.g. single-trial channel interpolation, ICA-based eye artifact removal). These data quality technicalities notwithstanding, we do agree with the reviewer that it would be helpful to provide some reassurance that a 0.1 Hz high-pass filter cannot falsely remove a slow-evolving prior effect. We therefore applied a 0.1 Hz high-pass filter to the continuous MBL data (Fig. R2), and, despite the expected increased noise level due to reduced artifact removal, it is clear that applying a 0.1 Hz high-pass filter to the MBL data does not reduce the sustained prior encoding effect (compare black traces in Fig. R2A vs. Fig. R2C). Meanwhile, as expected, applying a more severe 1 Hz high-pass filter does remove this effect (Fig. R2B). These analyses substantiate the point that, given the trial lengths in our task, a 0.1 Hz high-pass filter is too low to remove slow-building decision-related activity where it is present, as is the case in MBL. In turn, the fact that no sustained prior activity was observed in the CPP cannot be accounted for by application of the 0.1 Hz filter.

Since Reviewer 2 also had a query regarding the impact of high-pass filter settings, we have also included a comparison of filter settings for the CPP in the figure below. This shows that, in contrast to MBL, the application of a 0.1 Hz and 1 Hz high-pass filter to the CPP data does not affect the $|\text{prior}|$ encoding regression results (Fig. R2D-E).

Figure R2. A-C. Replication of the main analyses in Fig. 2E, Eq. 10, after applying a 0.1 Hz (A), 1 Hz (B) or no high-pass filter at all (C) to the motor beta-band lateralisation activity, extracted from continuous EEG data, in a subset of $N = 18$ participants (two participants were excluded due to excessive noise in the continuous data). The 1 Hz high-pass filter removes the slowly evolving decision-related changes in the signal captured by the sustained prior (Ψ) encoding effect (B), whereas the application of a 0.1 Hz high-pass filter to MBL data does not alter this effect (compare A to C). **D-E.** Regression coefficients for an extended version of Eq. 12 (Eq. S3) including both $|\Psi|$ and $|\Delta\Psi|$ as regressors after applying a 0.1 Hz (D) or 1 Hz high-pass filter (E). The filtering settings did not substantially change the (lack of) estimated prior encoding effects.

Finally, I do not understand how the authors can assert that CPP is not doing any integration when one looks at Figure 3A, where the EEG amplitude for the first few samples is clearly smaller than the last few.

> This is a key point, and we are glad to have a chance to clarify one of our main results. The CPP has previously been described in decision-making tasks with continuous sensory input

that is accumulated over durations of typically less than 2 seconds (see e.g. O'Connell et al. 2012). In these contexts, centroparietal signals show a growing positivity that peaks at the time of action and scales with the strength of an unsigned DV (i.e. the amplitude tracks the absolute cumulative sum of evidence). We reasoned that, if the CPP were to behave the same way in our longer lasting, discrete-token task, the increasing positivity observed in some of the channels would track the evolving unsigned decision variable, which in our task is quantified as $|\Psi|$. Our analysis in Fig. 3B directly tests this. If it were the case that channels exhibiting a positive trend (i.e. the ones where later samples have a greater amplitude than early samples, as the reviewer mentions) encode an integrated decision variable, we would expect this amplitude to scale with the strength of that decision variable. That is, samples where beliefs are stronger (i.e. $|\Psi|$ is higher) should exhibit higher amplitudes. The fact that we find no such effect indicates that the positive drift seen in some channels is unrelated to the evolving state of a decision variable.

To show that our conclusions about $|\Psi|$ encoding do not depend on our electrode selection, we now show regression coefficients for various subsets of electrodes for the analysis depicted in Fig. 3B. No significant effects of $|\Psi|$ emerged in any subset of channels. (Fig R3, A-D). We further show that the scalp distribution of the slow-building positive component observed in some electrodes has predominantly frontal and central foci, suggesting that these positive-going slow potentials may emerge from different sources than the conventional centroparietal positivity (Fig R3F,I). Neither of the two observed clusters showing strong slow positive-going deflections showed significant prior encoding (Fig R3G,J), suggesting that they do not reflect an evidence-integrating decision variable in the same way as the CPP observed in continuous stimulus tasks. Although discovering the function of these slow drifting components is beyond the scope of the current study, we can offer some speculation regarding a potential role in encoding elapsed time, with such signals identified in frontal cortex in previous non-human primate studies (e.g. Genovesio et al. 2006). We have added a comment to this effect in the discussion (p.21, lines 574 onwards).

While we therefore conclude that the CPP amplitude does not reflect the state of a decision variable in this task, our results suggest that the magnitude of the token-evoked CPP potentials reflects the intermediary quantity of 'effective evidence' ($|\Delta\Psi|$) taking into account both the prior belief and new sensory observation, as we show in Figure 3E. And it is worth noting that, in this case, the topography of this effect does indeed match the classical CPP one (see Fig. 3D).

We have incorporated Figure R3 in the supplementary material (new Figure S4) and added some comments to make our reasoning clearer in the results & discussion sections.

Figure R3. A-D) Regression coefficients for Eq. 11 in various electrode sets. Panel A reproduces Fig 3C in the manuscript, and panels B to D include smaller subsets of the original broad electrode selection. No significant prior ($|\Psi|$) effects emerged anywhere on the scalp. **E,H)** Whole-trial ERPs, for central (E) and frontal (H) subsets of channels showing a strong positive drift from trial onset to trial end. **F,I)** Difference topographies between trial end (sample 10, [4-4.4s]) and trial start (Mask, [0-0.4s]). Positive values indicate channels with a positive-going drift, exhibiting higher amplitudes towards the end of the trial. **G,J)** Regression coefficients for the subset of central (G) and frontal (J) positive-going channels. Neither of these two subsets showed significant encoding of prior beliefs, indicating that the signal amplitude did not scale with the strength of prior beliefs.

On the second part of the conclusions, the evidence for CPP feeding into MBL is relatively small. It looks like that this conclusion comes mostly from the analysis shown in Figure 4F, but the effect seems to disappear when the analysis is split between early and late samples (right panel). Altogether, the conclusions are appealing, but the evidence is not very convincing. We would like to clarify that in the panel mentioned by the reviewer where the results were split by early/late samples (Figure 4F right) it is a difference *between* early/late samples that we tested for and found to be nonsignificant, not tests for the presence of an effect in *either* the early or late ones, to which the reviewer refers. The effect therefore does not disappear, and we have clarified the text to make sure to avoid any such misunderstanding.

We would also like to clarify that panel F was not the only source of evidence for this point: the fact that residual fluctuations in token-evoked CPP amplitude have a significant impact on that token's weight on choice, as reported in Figure 4A-B, also lends some (indirect) support to our claims in the sense that, in order for that effect to be present, residual fluctuations in CPP amplitude must somehow be translated into a downstream effect in action-generating areas. We have added some nuance to the results section to make our reasoning clearer (p.16, lines 436 onwards).

Third, informed by the conclusions of the previous paper (Murphy et al., 2021), I was expecting that the authors would test new predictions. For instance, one could have tested different values of the hazard rate (including a baseline condition where this rate is zero), to check whether CPP would be sensitive to the hazard rate. One could have added a secondary task to ask participants if they detected a change of distribution within a trial, and analyze the EEG signals separately depending on whether participants did indeed detect a change. Again, I think the analyzes presented here are well performed, but the project lacks ambition.

> Previous work has shown that the normative model we work with here is able to capture behaviour well in a variety of volatility conditions (Glaze et al. 2015, Murphy et al. 2021). While it would have been interesting to replicate our findings across several levels of volatility, and the question of whether and how the subjects' subjective impressions accorded with the EEG signal amplitudes is interesting, our aim here was to gain new insights into the cognitive processes underpinning normative accumulation using one particular volatility value as a paradigmatic example. In relation to this aim, our key results were 1) the lack of $|\Psi|$ encoding across discrete samples in centroparietal signals, which stands in contrast to canonical CPP activity in continuous evidence tasks (Fig. 3B), 2) the fact that token-evoked CPP potentials encode momentary belief updates (Fig. 3C-D), a quantity directly derived from the normative model that captures a crucial intermediary step between sensory encoding and motor preparation, and 3) the direct evidence that single-trial fluctuations in the CPP amplitude have an impact on behaviour (Fig. 4A). None of these had been previously established, and provide novel insights into the neural underpinnings of normative evidence accumulation and the flexible functional role of the CPP in decision formation. We have edited our manuscript to make the significance of our study clearer (see e.g. p.20, lines 555 onwards).

Reviewer #2 (Remarks to the Author):

In this study, Parés-Pujolràs and colleagues investigate the functional role of two important neural signatures found in evidence accumulation and decision-making. They employed an experimental paradigm designed to be able to disentangle their roles, in particular, with a lower time scale allowing them to study what happens at each sample of evidence delivered.

Using an elegant combination of computational modeling and electrophysiology, it was found that the centroparietal positivity signal relates to belief update sample by sample, rather than an overall decision variable evolution throughout the samples. Via a smart use of the "residuals analysis approach", another key finding is linking the fluctuations in the centroparietal encoding of belief updates to the variability in motor beta lateralisation encoding of the decision variable.

Please note that my expertise lies primarily in decision-making and belief updating, rather than EEG.

The paper is particularly well written and very clear, because the authors describe their reasoning motivating each piece of analysis. Nevertheless, I have some comments:

> We would like to thank the reviewer for their careful reading of our manuscript and constructive comments.

General comments

1. Have the authors have tried to apply a 1 Hz high pass filter for the main analysis as in previous work rather than the 0.1 Hz low pass filter, to support their discussion point that it matters for identifying any sustained activity across the samples? (not only slow drifts but relevant activity?) How does Fig S3 contribute to this point?

> Our reasoning on sustained activity was that, based on previous CPP literature, any sustained activity related to the evolving decision variable would be manifested as slow changes across samples. Translated to our regression-based approach, this would amount to samples associated with stronger priors ($|\Psi|$) being linked to higher positive signal amplitudes; and specifically, to sustained non-zero effects of $|\Psi|$, including at sample onset, similar to what we observed for MBL. Figure S3 in our original supplementary materials reproduces the CPP analysis in Figure 3A (whole-trial ERPs) and C (CPP encoding of effective evidence) on 1 Hz high-pass filtered data. Since the original main analysis in Figure 3B (CPP multiple regressions with $|\Psi|$, $|\text{LLR}|$ and $|\text{LLR}|*\text{surprise}$ terms) did not find any sustained $|\Psi|$ effect and the 1 Hz high-pass filter effectively removes slowly evolving activity of the kind we were expecting sustained decision-related activity to be, we did not originally repeat that specific analysis on 1 Hz high-pass filtered data. However, following the reviewer's comment we have now done so, thus reproducing results in Figure 3B but now with 1 Hz high-pass filtered data, along with the effective evidence regression in Figure 3C extended here to include the $|\Psi|$ regressor (Fig. R4A-B below). As expected, no significant sustained effects of $|\Psi|$ were found in these analyses. To further illustrate the effect of a 1 Hz high-pass filter on slowly-evolving signals, we have additionally re-run the analysis in Fig. 2E (MBL multiple regressions with Ψ and $\Delta\Psi$ terms) after applying a 1 Hz high-pass filter to motor beta lateralisation. This effectively removes the slowly evolving changes in beta lateralisation that we already show reflect the evolving belief, and therefore no sustained effect of prior can be identified anymore (Fig. R4.C below). See also Fig. R2 above, where, in response to a query by reviewer 1, we compare the effects of various high-pass filter settings on the MBL signal where a slowly evolving signal is present, versus the CPP where it is absent.

Figure R4. A) Repeat of the main analyses in Fig. 3B but with a 1 Hz high-pass filter applied. No sustained effects of prior ($|\Psi|$) emerge, as in main analysis. **B)** Regression coefficients for

an extended version of Eq. 12 from the main text (Eq. S3) including both prior ($|\Psi|$) and effective evidence ($|\Delta\Psi|$) as regressors (and as in **A**, applied to 1 Hz high-pass filtered data). Again, no significant encoding of prior emerged. **C**) Replication of the main analyses in Fig. 2E after applying a 1 Hz high-pass filter to the motor beta-band activity. This effectively removes slowly evolving changes in the signal (i.e. increasing lateralisation with increasing belief strength), and therefore the effect of prior disappears.

An alternative way in which the CPP might encode the state of a DV in a non-sustained way would be in transient token-evoked activity. For example, it might have been the case that stronger beliefs (higher $|\Psi|$ values) were associated with larger evoked responses. However, we didn't find that to be the case in either the main analysis (0.1 Hz high-pass; Fig. 3B) or the supplementary ones (1 Hz high-pass; Fig. R4A,B above).

2. Have the authors analysed separately the trials that actually follow a true change point (not only the change points occurring during the sequence of samples)? Or is it not relevant because we cannot pinpoint exactly at which sample/which trial the participant picks up on the change point that occurred? (I would have expected the effect of effective evidence to be stronger after a true change point as compared to all sampled pooled).

>We thank the reviewer for this helpful suggestion to analyse potential differences in effective evidence at change point compared to non-change point samples. (To avoid possible confusion: by change point we specifically mean actual changes in the generative distribution, as opposed to consecutive samples that happen to appear on opposite sides of the screen but are produced by the same underlying generative distribution). Based on the normative model, the most extreme effective evidence values will occur in cases where both 1) prior beliefs are strong, and 2) new tokens provide evidence that goes strongly against those beliefs. These two circumstances are more likely to co-occur with a change point (though not always), thus resulting in higher average effective evidence values carried by samples coinciding with change points. To illustrate that point, and following the reviewer's comment, we have now grouped all samples coinciding with a change point (CP) and those that didn't (noCP), and plotted the average absolute effective evidence ($|\Delta\Psi|$) and CPP traces for those subsets of trials. As expected, change-point samples were associated with higher $|\Delta\Psi|$ values, and a correspondingly higher CPP positivity was elicited by these samples (see Fig. R5). We now report these findings in a new supplementary Figure S6.

This new analysis is similar in spirit to our existing analysis shown in Fig. 3E, in that it provides supporting evidence for the encoding of belief updates through standard waveform averaging for selected trials adhering to exemplary cases, outside of the main regression analyses. We note that the average $|\Delta\Psi|$ values at change points are smaller for this new analysis than the most extreme ones observed in the existing analysis (Fig. 3E), which illustrates that not all change points are associated with high $|\Delta\Psi|$ values. This is not surprising since, by design, not all change-points on our task will immediately produce evidence samples that evoke a large belief update (i.e. in cases where the new evidence happens to be weak, or the new evidence is consistent with the existing belief despite the change in generative distribution). This highlights the fact that analyses based only on the presence or absence of change-points will have limited sensitivity when applied to our design, in contrast to the regression-based approach that we employ in most analyses which captures the full range of variability in the factors driving belief updates.

Figure R5. A) EEG amplitude at centroparietal channels following samples at a change point (CP) or in the absence of a change point (noCP). Change point samples evoked higher amplitude CPPs than noCP ones ($p = 0.018$, two-tailed permutation test). Data are baselined at sample onset to remove spurious drift-related baseline shifts. *Asterisks along the bottom indicate significant cluster periods for tests averaging over the highlighted electrodes. **B)** Mean (\pm s.e.m.) absolute effective evidence ($|\Delta\Psi|$) associated with change point (CP) were significantly higher ($p < 0.001$) than those associated with no-change point (noCP) samples.

3. The residual effect for motor preparation did not vary between early and late samples, unlike the MBL effect; however, there seems to be a small difference later after sample presentation (Fig 4F), do the authors think it could be a meaningful late effect?

>The reviewer highlights an interesting deflection present in the residual effects of late samples 0.5s after sample presentation. It is tempting to speculate that this deflection might be related to the link between the CPP, MBL and the impact of residual motor fluctuations on behaviour in later samples. We reanalysed the residuals data testing specifically for differences in the late deflection and we found no significant effects (Fig. R6). Therefore, we would be reluctant to speculate at this stage - especially since we do not have a strong hypothesis regarding the potential functional role of this late component and effects were not found to be significant.

We would further like to thank the reviewer because working on this analysis helped us to identify a small timing issue in some of our participants. While reanalysing these data we noticed that, due to a technical issue, 7/20 participants had a 14ms shorter stimulus onset asynchrony than expected, which resulted in a minor misalignment of the data in our sample-wise regression-based analyses (including those leveraging MBL or CPP residuals) that was strongest for later samples. We have now corrected all of the analyses and corresponding statistics in the manuscript. None of our results change after the alignment correction.

Figure R6. Standardised regression coefficients (mean \pm s.e.m.) of Eq. 17, showing that variability in centroparietal responses influences motor beta lateralisation updates (as in Fig. 4F). Shaded areas indicate cluster test periods, either testing for differences between Early vs. late samples (A) or significant deviations from zero in early (B) or late (C) samples respectively. Shaded areas indicate the cluster test period [0.5-0.7s].

4. Do the authors think that the CPP results would generalise to tasks where mappings between stimulus and response are not known beforehand, or can change before the response is provided?

> The reviewer raises a very interesting question. Given that in conventional, continuous stimulus tasks the CPP has been shown to be present both when response mappings are unknown (Twomey et al. 2016) and when no overt movement is required (O’Connell et al. 2012), we would a priori predict that in the current task it should also behave the same way. However, given that at this stage it is unclear how exactly the non-linear normative transform is implemented in the brain and which neural circuits the CPP receives input from, this is an hypothesis that would need to be tested. Furthermore, our results indeed beg the question of where an abstract, effector-independent decision variable would be encoded in the context of a discrete token task like this one but where mappings are unknown. While in conventional continuous evidence tasks the CPP activity is consistent with an integrated, effector-independent decision variable, it is unclear what signal might maintain a sustained representation of a DV in a discrete token task like ours if neither the CPP nor MBL take on that role. We have added a comment in the discussion about this (p.24, lines 652 onwards).

5. Even though the Results section describe which findings replicate the MEG results of Murphy et al., it would be interesting to have a broader discussion of how much overlap/whether any differences, also, are seen, in the Discussion section or Supplement

>Thank you - we have added a broader discussion on how our findings replicate and add to previous MEG findings in the discussion section (see p.20, lines 555 onwards).

“While previous MEG work (Murphy et al. 2021) was able to trace the encoding of objective sensory evidence through visual cortical areas and, at the far end of the pathway, a motor plan reflecting the evolution of the normative decision variable across tokens, no correlates of the intermediate neural processing steps required to transform the former to the latter had been identified. Our finding that the CPP uniquely encodes this critical normatively-scaled quantity therefore adds a missing link to the chain of events leading from objective stimulus information to a normative decision variable ”

6. It would be useful that the introduction and results section better describe the non linearity aspect between the posterior belief and the prior of the next sample for readers unfamiliar with the volatility model of Glaze et al.

>Thank you - we have added a comment explaining the kind of non-linearity the normative model relies on with an inset and accompanying caption in Fig. 1B, which illustrates how it changes across various levels of environmental volatility.

Minor comments

7. The notion of stationary stimulus: RDK may not be characterised as stationary but indeed too fast as the authors argue; perhaps another word that 'stationary' would fit better?

>We used the term "stationary" in the statistical sense, and in contraposition to "volatile". That is, we use stationary to refer to an environment where the generative distribution does not change over trial time (i.e. hazard rate = 0). We also comment on the "too fast" nature of the RDK as a separate aspect and to contrast it with the discrete tokens used in our task - we have clarified this in the text (p.4, line 120).

8. Could the authors specify participants' compensation (nature and amount)

>Participants were paid 10€/hour - we have added that to the methods section of the manuscript (p.26, line 702).

9. In the introduction, could the authors develop the biases they refer to in "CPP subject to additional task-dependent biases (line 100)"

>Thank you - we have clarified this in the text (p.4, lines 94 onwards).

Reviewer #3 (Remarks to the Author):

Pares-Pujolras and colleagues examine electrophysiological candidates for a decision variable in a complex task in which momentary evidence must be updated dynamically in accordance with the possibility that the correct response can change over time. The authors show that motor beta lateralization evolves slowly and matched the computational profile of a normative decision variable, whereas parietal event triggered responses (centroparietal positivity) better matched signals for moment to moment belief adjustments, rather than an accumulated evidence signal. The authors show that both signals relate to behavior in ways that support these computational categorizations and furthermore, that residual fluctuations in CPP strength covary with the degree to which new observations lead to changes in beta lateralization.

Overall, I found this paper to be interesting, timely, and well communicated. Nonetheless, I have a few questions and concerns about the manuscript in its current form.

>We would like to thank the reviewer for their careful reading of our manuscript and constructive comments.

One question I have is whether participants make horizontal saccades when seeing the most disconfirmatory information, and whether any such eye-movements might contaminate the overarching results. It is clear that successive stimuli that alternate between left and right sides of the screen would drive large belief updates in the model and likely be interpreted as changepoints. But I can't tell from the methods how big the arc is, and to what extent participants might break fixation to gaze at new stimuli rather than just viewing them in periphery. CPP doesn't seem like the sort of ERP that would be elicited by eye movements, but given that some of the key variables almost certainly relate to alternative lateral screen positions, I think it is worth considering eye movements carefully.

>We have now included details on stimulus eccentricity in the methods section (p.26, line 722). Participants were instructed to maintain fixation at a central point throughout trials, and we used Independent Component Analysis (ICA) to remove both eye blink and horizontal eye movement artifacts from the signal. This should have removed (or at least heavily reduced) the potential for signal contamination, but it might not have removed saccade-related neural

potentials (Yuval-Greenberg et al. 2008). Given that we recorded eye movements using an EyeLink eye-tracker during this task, we have further looked into how often participants broke fixation and performed some related additional control analyses.

We used a conservative procedure for saccade detection based on velocity (30 deg/s), acceleration (8000 deg/s²), and a minimum displacement of 0.5 degrees, following default EyeLink criteria and previous work (Engbert & Kliegl, 2003; Jackson et al. 2008). This ensured we also captured microsaccades (see Fig. R7A).

There was variability in saccade behaviour across participants. On average, participants broke strict fixation on approximately 20% of the samples (in the interval between current-sample onset and next-sample onset). To test whether the corresponding saccades might have contaminated our key results, we carried out two additional analyses. First, we tested whether our key variables of interest ($|\text{LLR}|$ and $|\Delta\Psi|$) significantly predicted the presence/absence of saccades using two logistic regressions with each of these variables as a regressor, and we then compared the goodness of fit of both regressions using adjusted R-squared values. We found that, while some participants showed a significant association with only $|\text{LLR}|$ (3/20), only $|\Delta\Psi|$ (6/20) or both (4/20), saccade behaviour was slightly better captured by $|\text{LLR}|$ at the group level (Fig. R7B). This was also the case when we restricted our analysis to participants showing a significant association with either one of the two regressors (Fig. R7B). $|\text{LLR}|$ maps onto the distance away from the vertical meridian centered on our fixation point, and the association observed in some participants indicates that they were more likely to saccade when more “extreme” tokens appearing toward the far left or right edges of the semicircle were presented. The fact that the $|\text{LLR}|$ regression gave a slightly better fit than the $|\Delta\Psi|$ one indicates that it’s unlikely that our main results in Fig. 3D (i.e. where $|\Delta\Psi|$ accounts for more variance in single-trial CPP amplitude than $|\text{LLR}|$) could be driven by overlapping saccade-related potentials. However, to further test that this was not the case we repeated all analyses in Fig. 3B to D excluding all samples where either a saccade was detected or EyeLink data were poor quality. This resulted in the exclusion of an average of 30% of samples across participants from the regression analysis.

We found that the results were unchanged by this exclusion (Fig. R7D-E), with CPP data still being best explained by the $|\Delta\Psi|$ model (Fig. R7F). These analyses suggest that our results were not driven by potentially overlapping saccade-related potentials. We have included these results as Supplementary Figure S5 in the revised paper.

Figure R7. **A.** Example trial with 3 saccades detected based on velocity (top), acceleration (middle) and degrees of displacement (bottom). Vertical dashed lines indicate sample onset times, and horizontal red dashed lines indicate thresholds for saccade detection. **B-C.** Mean (\pm sem) goodness of fit (adjusted R squared) of logistic regressions predicting saccade presence or absence based on either $|LLR|$ or $|\Delta\Psi|$. While some participants showed significant associations (see inset; dashed line indicates $p = 0.05$), saccades were marginally better explained by a sample's objective evidence ($|LLR|$, which maps onto distance from the vertical meridian of the screen) rather than its effective evidence ($|\Delta\Psi|$) in both the full participant sample (**B**) and the subset of participants showing significant associations with either one or both regressors (**C**). This difference was not significant ($p = 0.329$ (**B**), $p = 0.310$ (**C**); paired t-test of adjusted R squares). **D-F.** Replication of results in Figure 3B-D of the main manuscript after excluding samples with saccades from analysis.

How distinct are $|LLR|$ and $|LLR|*surprise$? I would think that these things are pretty highly correlated, and only distinguished by some pretty specific task situations... It sort of seems like what the authors are trying to get at with the latter regressor is the idea that the response to a given evidence will be stronger if that evidence is on the opposite side of the screen as the decision variable up to that point in time. As a complement to the regression analysis, I could imagine just looking at this directly, but plotting the magnitude of CPP as a function of evidence location separately for cases where preceding decision variable was on either same or opposite side of screen.

> The $|LLR|$ and its interaction with surprise are weakly negatively correlated when both are normalized (z-scored) as per our approach for regression analyses (see Fig. R8 below). Given that surprise is computed through an interaction of prior beliefs with new evidence, as the reviewer alludes to, large $|LLR|$ values are only accompanied by large surprise values in some situations - namely, when the sign of the large (signed) LLR does not match the sign of a strongly held prior belief (Ψ); while conversely, large $|LLR|$ values will be accompanied by very small surprise values when the sign of the LLR matches the sign of a strongly held prior. It is

this dissociation between $|\text{LLR}|$ and surprise that decorrelates the $|\text{LLR}|$ and $|\text{LLR}|*\text{surprise}$ terms.

In a similar vein to the analysis suggested by the reviewer, we complemented our regression analyses by plotting specific subsets of samples in Figure 3E. We took a subset of samples with similarly high $|\text{LLR}|$ values and split them according to their effective evidence ($|\Delta\Psi|$) values. We focussed on the highest $|\text{LLR}|$ values because these are the ones that show the highest variability in the $|\Delta\Psi|$ values associated with them, or in other words, in how “surprising” or “unsurprising” they can be. The highest $|\Delta\Psi|$ bin in our Figure 3E analysis corresponds to the occasions where the $|\text{LLR}|*\text{surprise}$ interaction term would be largest: such extreme values can only occur in cases where prior beliefs were strong, and new samples that strongly oppose current beliefs are displayed. In addition to that original analysis, we have now followed the reviewer’s suggestion to plot average CPP traces sorting tokens according to whether they appeared at a side consistent or inconsistent with the decision variable up to that point in time (Fig. R9). We observe that, while inconsistent evidence elicits slightly higher belief updates, the average difference is very small and the numerically higher CPP amplitudes do not reach significance. For a similar supplementary analysis, please see Fig. R4 in response to reviewer #2’s query above, where we sorted tokens as a function of whether or not a token coincided with a change point in the generative distribution. Average $|\Delta\Psi|$ differences between bins are bigger in that example, and so are the observed differences in the CPP.

Figure R8. Pearson correlations between relevant variables, computed for one example subject across samples [2-10].

Figure R9. A) EEG amplitude at centroparietal channels following tokens consistent or inconsistent with prior beliefs (i.e. LLR sign is either the same or different from Ψ sign). Data are baselined at sample onset to remove spurious drift-related baseline shifts. **B)** Mean (\pm s.e.m.) absolute effective evidence ($|\Delta\Psi|$) associated with consistent or inconsistent samples.

While I definitely understand the a-priori focus on the CPP and beta lateralization, I was left wondering whether these signals are unique, or whether there might be other similar signals that would emerge if the same regression analysis were examined across the sensor space.

> All CPP regression analyses were run for all 128 electrodes separately, and the standardised beta coefficients of those whole-scalp results are described in the topographies in Fig. 3B and C. For our statistical tests we did however average the results over a broad cluster of a priori-selected centroparietal electrodes to increase our power. The topography plots of the statistical results suggest a widespread encoding of objective evidence ($|\text{LLR}|$), with two foci of particularly strong encoding - one more frontal and one more posterior. The more posterior focus was revealed to be the most selectively associated with the transformed effective evidence quantity (Fig. 3D - note that this topography shows the difference in the adjusted R squares for regressions against only $|\text{LLR}|$ (Objective evidence) vs. only $|\Delta\Psi|$ (Effective evidence), which again were performed on all sensors separately). We have further analysed various separate subsets of electrodes (see Fig. R2 in response to Reviewer #1 above, and included as the new Figure S4), none of which showed qualitatively different effects from the ones reported in our original analyses.

In another set of scalp-wide analyses, we explored the possibility that a pure “surprise” signal (perhaps corresponding to the classic P3a ERP, which has a frontal topography) could be driving some of our observed effects, in particular the frontal cluster described above. Our findings (see Figure S7) do not support this possibility, though we acknowledge that study designs that orthogonalise different forms of surprise and manipulate them independently (e.g. Nassar et al., 2019) will be better positioned to address such questions.

Regarding the beta band results, following the reviewer’s comment we repeated the analysis for all electrodes and plot the effect topographies below (Figure R10). Prior and effective evidence effects were encoded in motor areas, as reported in the main manuscript, but also in a lateralized manner over occipital electrodes. This complements previous work in MEG, which identified encoding of prior beliefs and normatively scaled evidence in visual cortical areas (Murphy et al. 2021) in the alpha band. These analyses are promising and suggest that indeed oscillatory effects in our dataset extend beyond the motor cortex. However, a thorough investigation of encoding of normative variables in posterior EEG signals would require more extensive work (in particular, additional analyses to disambiguate whether the prior and

effective evidence effects we show over posterior channels in Figure R10 are truly due to encoding of these variables or if these effects may instead be explained by multicollinearity with other decision-relevant variables like objective evidence). Given our a priori, hypothesis-driven interest specifically in the CPP and MBL, and the fact that similar effects with posterior oscillatory signals have already been investigated in detail elsewhere (Murphy et al., 2021), we believe that a systematic investigation of these effects is beyond the scope of the current paper.

Figure R10. **A** Topographies illustrate whole-scalp regression coefficients for beta band power effects [0.25-0.3s] post-sample, averaged over samples 2-10, for a regression including Ψ and $\Delta\Psi$ regression in Eq. 10. **B.** Standardised regression coefficients (mean \pm sem) for the left- and right-hemisphere electrodes over the motor cortex used to compute the MBL effects reported in the main manuscript and highlighted in the inset. **C.** Standardised regression coefficients (mean \pm sem) for the left- and right-hemisphere occipital electrodes highlighted in the inset.

If I understand the model correctly, the sigmoid is essentially modeling the transition function of the task, which when applied in the space of log probability ratios, leads to floor and ceiling corresponding to the minimum probability for either state (due to the hazard rate). Is that correct? And if so, are the differences in updating from trial to trial mainly due to whether the evidence pushes the DV toward an extreme (in which case it gets pushed back by the transition function) versus towards complete uncertainty (in which case it is totally unaffected by the transition function)? Assuming my understanding is correct, I guess I wonder whether these processes are expected to happen at the same time... because it seems plausible to me that one might update their decision variable with respect to the evidence, and hold off on applying the transition function, perhaps until observing the next piece of evidence. Basically, my point here is just that I think that the authors are making some assumptions about when different stages of computation are happening with their analyses, and I think it would be really good for them to make those assumptions clear and justify them where possible.

> The reviewer raises an important question and one that we considered as we worked on this project. The reviewer's understanding is correct: the non-linear transfer function in the normative model determines how the posterior belief after accumulating a new evidence sample (x-axis) is non-linearly transformed to the prior belief for the next sample (y-axis) as a function of hazard rate, effectively "reining in" extreme posteriors particularly when the hazard rate is assumed to be high. If evidence pushes the DV "too high" ("too high" being defined by the hazard rate), the non-linearity pulls it back to a more uncertain level that is open to the possibility of change points. By contrast, when evidence is weak and/or posteriors are close to zero (where the function is quasi-linear with a slope close to 1), the effect of the

transformation is smaller. Compared to perfect evidence accumulation, this process makes it easier for new evidence to change the sign of the belief and thereby promotes more flexible decision-making in changing environments. The specific magnitude of belief updating in response to a given sample is jointly determined by the strengths and directions of both prior belief and new evidence, which by design (resulting from our choice of task hazard rate combined with generative signal-to-noise ratio) will all vary considerably across trials and samples on our task. In other words, variability in the magnitude of belief updating is subject to a number of different influences on our task that conspire to explore the full range of belief updating dynamics in the normative model.

While the non-linear transformation in the normative model is implemented, as the reviewer says, *after* a new sample of evidence has been accumulated (that is, it is the posterior and *not* the evidence itself that is transformed), we recognise that the brain may not produce behaviour resembling that of the model via precisely the same sequence of computations. Accordingly, we constructed our regression-based analyses to be free from assumptions about the order in which decision-related computations are carried out. If it were the case that the same sequence of computations prescribed by the normative model was followed by the brain (i.e. combination of prior with new evidence to compute posterior, followed at some point by transformation of posterior into prior for next sample), our analyses are capable of revealing that: In such a scenario, a signal reflecting the evolving decision variable would show strong and stable encoding of the prior present already at sample onset, later encoding of LLR (which sums with the prior to yield the untransformed DV, L), and later still sensitivity to surprise (which approximates the effect of implementing the normative non-linearity to yield the prior for the next sample). By contrast, the non-linearity could alternatively be applied *before* evidence is accumulated (i.e. by re-scaling the evidence before it's fed to the downstream accumulator, amounting to computation of our 'effective evidence' quantity $\Delta\Psi$), in which case LLR encoding and sensitivity to surprise should manifest at the same time. Our results mostly favour the latter scenario, given that the CPP displays early, *coincident* encoding of LLR and its interaction with surprise (Figure 3B) and, our analyses in Figure 4 suggest, feeds that information forward to MBL for accumulation. We do note an interesting hint at a more nuanced pattern in the downstream MBL signal itself, which shows some sign of delayed sensitivity to surprise relative to LLR (Figure 2D) consistent with sequential representation of first posterior and then transformed prior. Given that these encoding traces are noisy at the single participant level, there are associated challenges with comparing effect latencies, and our focus here has been on signal sensitivities to computational variables rather than detailed analysis of their latencies, we have opted to not pursue this analysis further in this manuscript. However, as the reviewer alludes to, we recognise that our current findings have some interesting implications for how the underlying decision-related computations may be carried out in the brain. We thank the reviewer for prompting us to consider this aspect of our results in more depth and have included some additional discussion of these points in the revised manuscript (p.23, lines 621 onwards).

Minor:

Topoplots in 3B could use a bit more labeling, I figured out color scheme eventually, but it was not obvious to me.

Thank you - we have added labels to the topoplots in Figure 3.

REVIEWER COMMENTS

Reviewer #1 (Remarks to the Author):

I would like to thank the authors for taking my comments, and those of the other reviewers, very seriously. I now understand better the novel contributions of the current manuscript. Regarding the issue of the slow positive drift that is visible in Figure 3A (last part of my second point), I find the response of the authors (top of page 6 of the reply to the reviewers) quite interesting, and I would encourage the authors to include a summary of this response in the main text (probably around line 329).

Thank you. We addressed the question of the positive drift just a few lines down (349-351) in the revised version of the manuscript, and included the related figure as a supplement.

Reviewer #2 (Remarks to the Author):

In their revision, the authors have now addressed the points raised earlier.

A minor final point would be to clarify the reasoning underlying the motivation for using EEG, as there is a degree of circularity in the reasoning: the authors say they need EEG to get the CPP as revealing intermediate steps of evidence accumulation; but also, that their findings help further characterise the nature of the CPP signal (specifically, as belief updating).

Thank you. We have ensured that the introduction of the paper avoids apparent circularity in using the CPP as a measure of something while also gaining insight into what it measures, by making it clear that though we assume a priori that it is giving us access to an intermediate stage, what we do not know and is tested here, is what is computed at that intermediate stage in this particular task context.

Reviewer #3 (Remarks to the Author):

The authors have fully addressed my concerns.

Thank you.